

# Water mass distributions and transports for the 2014 GEOVIDE cruise in the North Atlantic

Maribel I. García-Ibáñez[1], Fiz F. Pérez[1], Pascale Lherminier[2], Patricia Zunino[2], Paul Tréguer[3]

[1]Instituto de Investigaciones Marinas, IIM-CSIC, Eduardo Cabello 6, 36208 Vigo, Spain
[2]Ifremer, Univ. Brest, CNRS, IRD, Laboratoire d'Océanographie Physique et Spatiale (LOPS), IUEM, Plouzané, France
[3]Marine Environmental Sciences Laboratory (LEMAR, UMR 6539) at the European Institute for Marine Studies (IUEM), Université de Bretagne Occidentale, CNRS, F-29280 Plouzané, France

*Correspondence to*: Maribel I. García-Ibáñez (maribelgarcia@iim.csic.es)

**Abstract.** We present the distribution of water masses along the GEOTRACES-GA01 section during the GEOVIDE cruise,
which crossed the subpolar North Atlantic Ocean and the Labrador Sea in the summer of 2014. The water mass structure resulting from an extended Optimum MultiParameter (eOMP) analysis provides the framework for interpreting the observed distributions of trace elements and their isotopes. Central Waters and Subpolar Mode Waters (SPMW) dominated the upper part of the GEOTRACES-GA01 section in 2014. At intermediate depths, the dominant water mass was Labrador Sea Water, while the deep parts of the section were filled by Iceland–Scotland Overflow Water (ISOW) and North East Atlantic Deep
Water. We also evaluate the water mass volume transports across the 2014 OVIDE line (Portugal to Greenland section) by combining the water mass fractions resulting from the eOMP analysis with the absolute geostrophic velocity field estimated through a box inverse model. This allowed us to assess the relative contribution of each water mass to the transport across the section. Finally, we discuss the changes in the distribution and transport of water masses between the 2014 OVIDE line and the 2002–2010 mean state. At the upper and intermediate water levels, colder end-member of the water masses replaced the
warmer ones in 2014 with respect to 2002–2010, in agreement with the observed cooling of the surface and intermediate waters. Below 2000 dbar, ISOW increased its contribution in 2014 with respect to 2002–2010, increase related to the observed salinization since 2002. We also observed an increase in SPMW in the East Greenland Irminger Current in 2014 with respect to 2002–2010, which supports the recent deep convection events in the Irminger Sea. The assessment of the relative contribution of each water mass to the Atlantic Meridional Overturning Circulation (AMOC) across the OVIDE line allows
identifying the water masses involved in the increase in the AMOC intensity from 2002−2010 to 2014. The increase in the AMOC intensity is related to the increase in the northward transport of the Central Waters in its upper limb, and to the increase in the southward flow of SPMW of the Irminger Basin and ISOW in its lower limb.

**Keywords.** volume transports; OMP analysis; water masses; Meridional Overturning Circulation; Multivariate analysis; Mixing ratios.



## 1 Introduction

The 2014 GEOVIDE cruise consisted of two hydrographic sections: the seventh repetition of the OVIDE line from Lisbon (Portugal) to Cape Farewell (Greenland), and a section across the Labrador Sea from Cape Farewell to St. John's (Canada) (Fig. 1). The GEOVIDE cruise was the major French contribution to the Global GEOTRACES programme (official

GEOTRACES-GA01 section) aimed to achieve a three-dimensional distribution of the trace elements and their isotopes (TEIs) in the global ocean (SCOR Working Group, 2007). An important pre-requisite for achieving that goal is obtaining high quality hydrographic and tracer measurements that enable to trace back the origins, pathways and the processes governing the observed TEIs distributions. Water mass distributions are, therefore, a useful benchmark for these purposes (e.g., Jenkins et al., 2015). In this work, we qualitatively assess the water mass distribution along the 2014 GEOTRACES-GA01 section through an

extended Optimum MultiParameter (eOMP) analysis (Karstensen and Tomczak, 1998). We extend the study performed by García-Ibáñez et al. (2015) for the 2002–2010 OVIDE cruises, which successfully identified temporal variations and transformations of the water masses along the OVIDE line. As in that work, we also combine the water mass structure resulting from the eOMP analysis with the velocity field across the OVIDE line, obtaining water mass volume transports. The assessment of the water mass volume transports based on dilutions of "pure" water mases (OMP-based) provides insights on

the circulation features that are particularly useful for areas of complex currents and water mass transformation, as the subpolar North Atlantic (SPNA). Finally, we compare the water mass distribution and transport of the 2014 OVIDE line with the average water mass distribution and transport of the 2002–2010 OVIDE cruises, and link the observed changes to major changes in the formation and circulation of water masses in the SPNA. The differences in water mass distribution and transport between 2014 and 2002–2010 provides guidance in the interpretation of the observed distribution of TEIs.

## 2 Data and Methods

### 2.1 The hydrographic data

The GEOVIDE cruise (GEOTRACES-GA01 section) was conducted in June–July 2014 and consisted in 78 stations along the eastern SPNA and the Labrador Sea. Stations were labelled as Short (46 stations), Large (17 stations), XLarge (five stations), and Super stations (10 stations) depending on the number of sampled parameters and rosette casts. During the first cast in each

station, a classical rosette equipped with 22 Niskin bottles and CTD SBE911 equipped with a SBE-43 was deployed. This first station cast was used as reference for physical and chemical characterization of water masses. Discrete sampling for oxygen, nutrients and salinity was performed in all the 78 stations. Nutrient concentrations were measured by the continuous flow analyser (CFA), giving the concentration values in µM (µmol L$^{-1}$). We transformed measured nutrient concentrations in µM to µmol kg$^{-1}$ by diving by the density of the sample at 20ºC (measurement temperature). Accuracies were 0.001°C for

temperature, 0.002 for salinity, 2 µmol kg$^{-1}$ for oxygen and 0.2 µM for nutrients (nitrate and silicic acid). Upper water velocity was continuously measured with two ship-mounted Acoustic Doppler Current Profilers (ADCP; Ocean Surveyors).



For further reference, the vertical sections of potential temperature (θ), salinity, oxygen and silicic acid are shown in Fig. 2.

## 2.2 Hydrographic features and general circulation

The 2014 GEOTRACES-GA01 section crossed regions of water mass transformation and deep convection leading to water mass formation (e.g., Sarafanov et al., 2012; García-Ibáñez et al., 2015; Yashayaev and Loder, 2016). The North Atlantic

Current (NAC) carries warm and saline subtropical waters northwards, towards the northeastern Atlantic Ocean (Fig. 1). Air–sea interaction progressively reduces the temperature of the Central Waters transported by the NAC, being ultimately transformed into Subpolar Mode Water (SPMW) (McCartney and Talley, 1982; Tsuchiya et al., 1992). The continuous air–sea interaction along the NAC path leads to a continuous transition of varieties of Central Waters and SPMWs (McCartney and Talley, 1982; Pollard et al., 1996; Brambilla and Talley, 2008). The last stage of the transformation of SPMW is the

Labrador Sea Water (LSW), which is formed in the Labrador and Irminger Seas (e.g., Pickart et al., 2003; Fröb et al., 2016). After its formation, LSW enters the Deep Western Boundary Current (DWBC) (Bersch et al., 2007) (Fig. 1), where it joins the Denmark Strait Overflow Water (DSOW) and the Iceland–Scotland Overflow Water (ISOW) (Rudels et al., 2002; Tanhua et al., 2008). DSOW forms after the deep waters of the Nordic Seas overflow the Greenland–Iceland sill and entrain Atlantic waters (SPMW and LSW) (Read, 2000; Yashayaev and Dickson, 2008). Cascading events of Polar Intermediate Water (PIW)

also affect DSOW in the Irminger Sea (Olsson et al., 2005; Tanhua et al., 2005, 2008; Falina et al., 2012). ISOW forms after the Norwegian Sea waters overflow the Iceland–Scotland sills and entrain SPMW and LSW (van Aken and de Boer, 1995; Dickson et al., 2002; Fogelqvist et al., 2003). Then, ISOW flows southwards in the Iceland Basin along the eastern flank of the Reykjanes Ridge. Through the journey of ISOW in the Iceland Basin, entrainment events lead to the formation of the North East Atlantic Deep Water (NEADW) (van Aken, 2000). NEADW recirculates in the West European Basin (Fig. 1) and mixes

with the surrounding waters, including the Antarctic Bottom Water (van Aken and Becker, 1996), which results in the thermohaline properties of NEADW can be approximated as a line (Saunders, 1986; Mantyla, 1994). The waters of the GEOTRACES-GA01 section are also influenced by the saline Mediterranean Water (MW), and the relatively fresh Subarctic Intermediate Water (SAIW). MW enters the North Atlantic from the Mediterranean Sea after overflowing the Strait of Gibraltar (Ambar and Howe, 1979; Baringer and Price, 1997). SAIW originates in the western boundary of the subpolar gyre, i.e., the

Labrador Current (Fig. 1) (Arhan, 1990), by mixing between the warm saline waters of the NAC with the cold and fresher LSW (Iselin, 1936; Arhan, 1990; Read, 2000). The thermohaline properties of SAIW vary along its pathway towards the West European Basin, becoming saltier and warmer (Harvey and Arhan, 1988; Pollard et al., 2004).

The above described water mass formation processes occurring in the SPNA lead to the ventilation and renewal of the intermediate and deep ocean, and are the start process of the Atlantic Meridional Overturning Circulation (AMOC) (Kuhlbrodt

et al., 2007; Rhein et al., 2011; Sarafanov et al., 2012). The AMOC consists of two limbs: a warm northward-flowing upper limb mainly constituted by the NAC, and a colder southward-flowing lower limb. For the OVIDE line, the upper limb of the AMOC is constituted by the Central Waters, the SPMW of the Iceland Basin (IcSPMW), SAIW and MW; while the AMOC



lower limb is constituted by the SPMW of the Irminger Sea (IrSPMW), PIW, LSW, ISOW, DSOW and NEADW (García-Ibáñez et al., 2015).

**2.3 Extended Optimum MultiParameter (eOMP) analysis**

To solve the complicated water mass structure of the SPNA, we used an Optimum MultiParameter (OMP) analysis (Tomczak and Large, 1989). This technique has previously been used to describe in detail the origin, pathways and transformation of the main water masses in the SPNA (Tanhua et al., 2005; Álvarez et al., 2004, 2005; García-Ibáñez et al., 2015). Briefly, OMP analyses consider the properties of a given water sample to be the result of linear combinations of a finite number of water masses represented by the so-called Source Water Types (SWT) (Tomczak, 1999). In this study, SWTs were characterized by θ, salinity, oxygen, silicic acid and nitrate (Table 1). Once the SWTs and their physical and chemical properties are defined, the OMP analysis solves the mixing between SWTs by a least square method constrained to be positive definite, giving the fractions of each SWT ($X_i$) in each water sample:

$$d = G * X_i + r \qquad (1),$$

where d is the observed property in a water sample, G is the matrix containing the properties defining the SWTs, $X_i$ is the relative contributions of each SWT to the sample and r is the residual. An additional constrain is mass conservation, which ensures the contributions of all the SWTs sum to 100%:

$$\sum X_i = 1 + r \qquad (2).$$

In this study, we solved an extended OMP (eOMP) analysis (Karstensen and Tomczak, 1998), which accounts for the non-conservative behaviour of oxygen and nutrients by using Redfield-like stoichiometric ratios (R):

$$d = G * X_i + \Delta O_{2_{bio}}/R + r \qquad (3),$$

where R is 12 for silicic acid (Perez et al., 1993; Castro et al., 1998), and 10.5 for nitrate (Takahashi et al., 1985; Anderson and Sarmiento, 1994).

The eOMP analysis, then, permits assessing the biological influences on non-conservative properties and the changes in oxygen due to the remineralisation of the organic matter ($\Delta O_{2_{bio}}$). The separation between biological and mixing components allows a better quantification of the oxygen consumption (de la Paz et al., 2017), and can be a valuable tool when interpreting tracer distributions such as the TEIs.

To be able to compare the resulting water mass distribution and transport for OVIDE 2014 with the average of OVIDE 2002–2010 (García-Ibáñez et al., 2015), we used the same eOMP set up as in García-Ibáñez et al. (2015). We used 14 SWTs to solve the water mass structure along the GEOTRACES-GA01 section (Table 1, Fig. 3). The upper waters of the GEOTRACES-GA01 section were characterised by Central Waters and SPMW. The thermohaline range of the Central Waters was solved by defining two SWTs coinciding with extremes of the θ-S line defining the East North Atlantic Central Waters (ENACW):





ENACW of 16ºC (ENACW$_{16}$), whose θ-S characteristics match those from the warmer central waters of Pollard et al. (1996); and ENACW of 12ºC (ENACW$_{12}$), which represents the upper limit of ENACW defined by Harvey (1982). The change in temperature of SPMW along the NAC path cannot be accounted by the OMP analysis, since it is the result air-sea interaction (e.g., McCartney and Talley, 1982; Brambilla and Talley, 2008). This problem was solved by defining three SWTs to

characterize SPMW: SPMW of 8ºC (SPMW$_8$), SPMW of 7ºC (SPMW$_7$) and SPMW of the Irminger Basin (IrSPMW). SPMW$_7$ and SPMW$_8$ characterize the thermohaline range of SPMW in the Iceland Basin, being the θ-S of SPMW$_8$ those of SPMW formed within the Iceland Basin (Brambilla and Talley, 2008); and the θ-S of SPMW$_7$ those of SPMW found over the eastern flank of the Reykjanes Ridge (Thierry et al., 2008). The θ-S of IrSPMW characterize SPMW found in the Irminger Sea (Brambilla and Talley, 2008), and are close to those of the Irminger Sea Water (Krauss, 1995). The intermediate waters of the

GEOTRACES-GA01 section were characterised by LSW, MW and SAIW. The thermohaline properties of LSW were set as the characteristic values for the classical LSW as a long-term average (Lazier, 1973; Dickson et al., 1996). The properties of MW were taken from Wüst and Defant (1936) near Cape St. Vicente, where MW has its θ-S characteristics set after overflowing the Gibraltar Strait (Ambar and Howe, 1979; Baringer and Price, 1997). The thermohaline range of SAIW (4–7ºC and S < 34.9) was represented by two SWTs: SAIW of 6ºC (SAIW$_6$) and SAIW of 4ºC (SAIW$_4$), following the descriptions

of Bubnov (1968) and Harvey and Arhan (1988). Finally, the deep waters of the GEOTRACES-GA01 section were characterised by DSOW, ISOW and NEADW. The thermohaline properties of ISOW were defined as the ISOW properties after crossing the Iceland-Scotland sills defined by van Aken and Becker (1996). The thermohaline characteristics chosen for DSOW were selected from those found by Tanhua et al. (2005) after DSOW crossing the Greenland-Iceland sill. We also included PIW in the analysis to take into account the dense shelf water intrusions into DSOW. The thermohaline characteristics

selected for PIW are in agreement with those proposed by Malmberg (1972) and Rudels et al. (2002). NEADW was modelled by the definition of two SWTs equal to the end-points of the line defining the thermohaline properties of NEADW in the West European Basin (Saunders, 1986; Mantyla, 1994; Castro et al., 1998): upper NEADW (NEADW$_U$) and lower NEADW (NEADW$_L$).

In order to solve an over-determined system of linear mixing equations (Eq. 2 and 3), a maximum of four SWTs can be

considered simultaneously: one Eq. (3) per each variable defining the SWTs and five unknowns –four X$_i$s, and $\Delta O_{2_{bio}}$–. This inconvenience was solved by organizing the SWTs into 11 subsets or mixing groups (Fig. 3c), which were set based on the characteristics and/or dynamics of the SWTs in the SPNA. The mixing groups are vertically and horizontally sequenced, and share at least one SWT with the adjacent mixing groups to ensure water mass continuity (for more details about the eOMP set up see García-Ibáñez et al. (2015)). After obtaining the X$_i$s for each water sample, the X$_i$ of NEADW$_U$ were decomposed into

1.5% of MW, 18.4% of LSW, 29.5% of ISOW and 50.5% of NEADW$_L$ (van Aken, 2000; Álvarez et al., 2004; Carracedo et al., 2012; García-Ibáñez et al., 2015).

An important assumption of the methodology is that the physical and chemical characteristics of the SWTs are considered time invariant and equally affected by mixing; hence, changes in the properties of the water masses over time are reflected through water mass redistributions. To avoid changes in water mass properties due to air–sea interaction, we excluded the first 75 dbar





where non-conservative behaviour of temperature and salinity is expected. We also avoided solving the mixing where high percentages of fresh water are found, i.e., over the Greenland shelf, by restricting the analysis to water samples with salinity greater than 34.7 (Daniault et al., 2011).

We tested the robustness of the methodology through a Monte-Carlo simulation (Tanhua et al., 2005), where the physical and
chemical properties of both SWTs and water samples were randomly perturbed within the standard deviation of each parameter. This allowed assessing the sensitivity of the eOMP analysis to potential measurement errors and temporal variations in the physical and chemical properties defining the SWTs (Leffanue and Tomczak, 2004). A hundred Monte-Carlo simulations were performed and the eOMP equation system was solved for each of them. The standard deviation of the average $X_i$s (last column in Table 1) is lower than 0.05 (on per one basis), indicating that the methodology is robust. Additionally, our eOMP
analysis is consistent since its residuals (r in Eq. 3) lack a tendency with depth (Fig. S1), and the standard deviations of the residuals are slightly higher than the measurement errors (Table 1). Besides, the eOMP analysis is able to reproduce the measured values when substituting $X_i$s in Eq. (3), with a correlation coefficient ($r^2$, Table 1) higher than 0.993, indicating again the reliability of the eOMP analysis.

## 3 Results

### 3.1 Water mass distribution for 2014

The water mass distribution for the 2014 GEOTRACES-GA01 section (Fig. 4) was obtained through an eOMP analysis (Sect. 2.3).

The Central Waters ($ENACW_{16}$ + $ENACW_{12}$) occupy the upper eastern part of the 2014 GEOTRACES-GA01 section from the Iberian Peninsula until the Reykjanes Ridge (Fig. 4a,b), being $ENACW_{12}$ the dominant one. The contribution of $ENACW_{12}$
exceeds 90% in the first 500 dbar in the Iberian Abyssal Plain, following the maximum in θ and the minimum in silicic acid (Fig. 2a,d). The distribution of the Central Waters is associated with the NAC, which causes the thermohaline front (Fig. 2a,b) delimiting $ENACW_{12}$. The westward extension of $ENACW_{12}$ reflects its cyclonic circulation in the Iceland Basin and its southward flow over the eastern flank of the Reykjanes Ridge (Read, 2000; Pollard et al., 2004).

The cooler end-member of the Central Waters, $SPMW_8$, extends below $ENACW_{12}$ (Fig. 4c). Air–sea interaction processes
along the NAC transforms both $ENACW_{12}$ and $SPMW_8$ into $SPMW_7$ (Thierry et al., 2008; García-Ibáñez et al., 2015), which dominates the upper 1500 dbar of the Iceland Basin and above the Reykjanes Ridge (Fig. 4d). The distribution of $SPMW_7$ reflects the circulation of the NAC around the Reykjanes Ridge, from the Iceland Basin to the Irminger Sea (Brambilla and Talley, 2008). Further transformation of $SPMW_7$ through air–sea interaction along the path of the NAC leads the formation of IrSPMW in the Irminger Sea (Brambilla and Talley, 2008). The main core of IrSPMW is on the Greenland slope, from where
it extends eastwards until the Reykjanes Ridge (Fig. 4a). This distribution could indicate that the major region of formation of IrSPMW is the northwest of the Irminger Sea (Brambilla and Talley, 2008), from where the East Greenland Irminger Current



(EGIC) (Fig. 1) would transport it until the GEOTRACES-GA01 section and then to the Labrador Sea. Once in the Labrador Sea, IrSPMW could act as a precursor for the upper LSW (Pickart et al., 2003).

SAIW (SAIW$_6$ + SAIW$_4$) extends along the upper western part of the 2014 GEOTRACES-GA01 section, from the Labrador Sea to 20ºW (Fig. 4g,h), being SAIW$_6$ the main end-member. The maximum contributions of SAIW are in the surface layer

of the Labrador Sea, over the Greenland Slope and in the first 1000 dbar on the eastern side of the Reykjanes Ridge. This distribution reflects the formation and circulation of SAIW. After its formation in the Labrador Current (Arhan, 1990) (Fig. 1), SAIW subducts below the NAC that transports it to the Iceland Basin (Bubnov, 1968; Arhan, 1990; Read, 2000), where it mixes with SPMWs and Central Waters.

The dominant water mass in the 2014 GEOTRACES-GA01 section is LSW (Fig. 4e), which extends along the whole section.

The highest contribution of LSW is in the Labrador Sea, with LSW concentration reaching 100%. LSW fills the Labrador Sea from surface to almost the seafloor, with the higher concentrations found in the first 2000 dbar. The distribution of LSW in the Labrador Sea indicates recent ventilation (Kieke and Yashayaev, 2015), which is also true for the Irminger Sea (Yashayaev and Loder, 2016), where high concentrations of LSW extend from surface to about 2000 dbar. The high oxygen concentration found in the first 1500 dbar in both basins (Fig. 2c) corroborates the recent ventilation of the LSW layer in the Labrador and

Irminger Seas. The LSW concentration in the Irminger Sea is lower than in the Labrador Sea, being the first 1000 dbar of the Irminger Sea dominated by a mixing of IrSPMW and LSW. In our study, the thermohaline characteristics for LSW correspond to the long-term average classical LSW (Lazier, 1973; Dickson et al., 1996). The recent events of deep convection observed in the Irminger Sea (Yashayaev and Loder, 2016) led to the formation of LSW-like in the Irminger Sea, with characteristics closer to the IrSPMW end-member. High LSW concentrations are also found in the West European Basin between 1000 and

3000 dbar. The decrease in the contribution of LSW found over the Reykjanes Ridge, where waters are a mixture between LSW, SPMW$_7$ and ISOW, suggests strong mixing around and over the Reykjanes Ridge (Ferron et al., 2014). This strong mixing is also observed in the θ and salinity distributions (Fig. 2a,b), by a deepening of the isotherm of 6ºC and the isohaline of 35. Some authors termed as Icelandic Slope Water the admixture of ISOW and Atlantic waters found around and over the Reykjanes Ridge (e.g., Yashayaev et al., 2007), which in our study is represented by mixing group 4 (LSW, NEADW$_U$, ISOW

and SPMW$_7$; Fig. 3c).

The relatively high salinity and low oxygen concentration centred at 1500 dbar in the West European Basin (Fig. 2b,c) is the result of the northward flow of MW (Reid, 1979) (Fig. 4f). MW intersects the GEOTRACES-GA01 section over the Iberian shelf, between 500 and 2000 dbar, and spreads westwards until 20ºW. This westward extension may result from meddy transport (Arhan and King, 1995; Mazé et al., 1997) or may be associated with the Azores countercurrent (Carracedo et al.,

30  2014).

The western part of the 2014 GEOTRACES-GA01 section (west of 20ºW) below 2000 dbar is dominated by ISOW (Fig. 4b). The main core of ISOW is located on the eastern flank of the Reykjanes Ridge, reaching percentages greater than 90%. From this region, ISOW extends eastwards into the West European Basin, where it is progressively eroded and diluted into NEADW$_L$ (Fig. 4h). ISOW also fills the deep areas of the Irminger and Labrador Seas, with concentrations greater than 50%. This





distribution of ISOW is consistent with its circulation from the Iceland–Scotland sills, across the Iceland Basin along the eastern flank of the Reykjanes Ridge, crossing the Charlie–Gibbs Fracture Zone (CGFZ) (Dickson and Brown, 1994; Saunders, 2001), flowing cyclonically in the Irminger Sea (Sarafanov et al., 2012) and joining the DWBC (Rudels et al., 2002; Tanhua et al., 2008), and then flowing cyclonically in the Labrador Sea (Xu et al., 2010). The eastward extension of ISOW into the

West European Basin may be the result of some fractions of ISOW bypassing the CGFZ (Fig. 1), as reported by previous studies (e.g., Fleischmann et al., 2001; LeBel et al., 2008; Xu et al., 2010).

The bottom areas of the Labrador and Irminger Seas are occupied by DSOW (Fig. 4g). The DSOW distribution is coincident with a minimum of $\theta$ (<2°C), a maximum of oxygen and a relative minimum of silicic acid (Fig. 2a,c,d). The DSOW distribution supports the circulation scheme for DSOW, which overflows the Greenland–Iceland sill, joining the DWBC, and

then continues flowing within that current around the Labrador Sea. Percentages of up to 20% of PIW are present within the realm of DSOW (Fig. 4f), supporting the existence of entrainment of East Greenland shelf waters into DSOW (Tanhua et al., 2008; Falina et al., 2012; von Appen et al., 2014). PIW is also found in the continental shelves of Greenland and Canada, with the greatest contribution found on the Canadian shelf. The appearance of PIW on the Canadian shelf is in agreement with the exchange of Arctic waters occurring via the Canadian Arctic Archipelago, Baffin Bay and Davis Strait (Curry et al., 2014),

which then join the Labrador Current and intersect the GEOTRACES-GA01 section on the Canadian shelf (Fig. 1).

The dominant deep water in the West European Basin is $NEADW_L$ (Fig. 4h), which extends from 2000 dbar to the bottom. The location of $NEADW_L$ coincides with the higher concentrations of silicic acid measured in section (>20 µmol kg$^{-1}$; Fig. 2d), supporting the influence of Antarctic Bottom Water in $NEADW_L$ (van Aken and Becker, 1996).

### 3.2 Water mass volume transports for 2014

Water mass volume transports across the 2014 OVIDE line (Portugal to Greenland section) result from combining the water mass fractions from the eOMP analysis ($X_i$s) with the absolute geostrophic velocity field. The absolute geostrophic field orthogonal to the 2014 OVIDE line was estimated by a box inverse model, using the hydrological profiles measured at each station, and constrained by ADCP velocity measurements and by an overall mass balance of $1 \pm 3$ Sv northwards (Lherminier et al., 2007, 2010; Zunino et al., this issue).

To allow the combination of the $X_i$s with the absolute geostrophic velocity field, the $X_i$s were linearly interpolated in density coordinates to match the grid of the absolute geostrophic velocity. Then the $X_i$s were multiplied by the absolute geostrophic velocity field, obtaining the water mass volume transports orthogonal to the section. The resulting water mass volume transports were then integrated along the section to obtain the net water mass volume transports (represented in Sverdrups; 1 Sv = $10^6$ m$^3$ s$^{-1}$) (Fig. 5). Northward (southward) water mass volume transports are positive (negative). Errors were computed

by weighting the velocity errors by the $X_i$s.

We describe the water mass volume transports based on their contribution to the upper and lower limbs of the AMOC. Across the OVIDE line, the upper and lower limbs of the AMOC are separated by the isopycnal $\sigma_1 = 32.15$ kg m$^{-3}$ (Mercier et al., 2015; Zunino et al., this issue), where $\sigma_1$ refers to potential density referenced to 1000 dbar. The upper limb of the AMOC for





the 2014 OVIDE line is represented by the Central Waters (ENACW$_{16}$, ENACW$_{12}$ and SPMW$_8$; 13.8 ± 1.4 Sv), SPMW$_7$ (3.2 ± 2.1 Sv) and SAIW (SAIW$_6$ and SAIW$_4$; 1.4 ± 0.6 Sv) (Fig. 5). We also included the net northward transport of MW (0.7 ± 0.6 Sv) to the AMOC upper limb, since it contributes to the formation of intermediate waters in the SPNA (Reid, 1979, 1994). These flows altogether result in an AMOC upper limb of 19.0 ± 2.6 Sv for OVIDE 2014, which is in good agreement with the 18.7 ± 3.0 Sv for the intensity AMOC reported for OVIDE 2014 by Zunino et al. (this issue). The contributors of the upper limb of the AMOC agree with the subpolar (SAIW and SPMW$_7$) and subtropical (Central Waters) components of the AMOC at the OVIDE sections described by Desbruyères et al. (2013).

The AMOC lower limb at the 2014 OVIDE line is, then, constituted by the remainder water masses, i.e., IrSPMW (-10.0 ± 1.2 Sv), LSW (0.8 ± 1.7 Sv), ISOW (-4.8 ± 1.1 Sv), DSOW (-2.2 ± 0.4 Sv), PIW (-2.2 ± 0.2 Sv), and NEADW$_L$ (0.3 ± 1.7 Sv) (Fig. 5), resulting in a southward transport of -18.0 ± 2.9 Sv. The transport of PIW was split into that associated with its distribution in the first 2000 dbar, and that associated with its distribution deeper than 2000 dbar (samples assigned to mixing group 3; Fig. 3c), which was added to DSOW to agree with the cascading events occurring along DSOW pathway (Olsson et al., 2005; Tanhua et al., 2005, 2008; Falina et al., 2012).

## 4 Discussion

### 4.1 Differences in the water mass distribution between OVIDE 2014 and the average OVIDE 2002–2010

In this section, we describe and discuss the observed changes in the water mass distribution along the OVIDE line (Portugal to Greenland) between 2002–2010 and 2014 (Fig. 6). To obtain water mass proportions in the same grid for all the OVIDE sections, we interpolated the eOMP results for 2014 from this work and those for 2002–2010 from García-Ibáñez et al. (2015) to a common grid using a Delaunay triangulation. The selected grid was the sampling locations from the OVIDE 2010 cruise. The high mesoscale variability of the SPNA, with changes in the location of fronts and eddies (Zunino et al., this issue), leads to changes in the water mass proportions between 2002–2010 and 2014 in alternative patterns of increases and decreases (Fig. 6). However, some water masses show persistent and regionally localized changes linked to changes in the hydrography of the OVIDE section.

Since OMP analyses consider time invariant the properties characterizing the SWTs, the inter-annual variability in the water mass properties at formation is solved by OMP analyses through water mass redistributions. Therefore, the observed changes in the thermohaline properties between 2002–2010 and 2014 are reflected by the redistribution of water masses. Positive (negative) anomalies in the proportion of a water mass imply a gain (loss) in 2014 compared to 2002–2010.

In OVIDE 2014, the contribution of ENACW$_{16}$ east of 20°W decreased with respect to 2002–2010 (Fig. 6a), while ENACW$_{12}$ increased (Fig. 6b); and IrSPMW replaced SPMW$_7$ over the Reykjanes Ridge (Fig. 6a,d). The replacement of the warmer end-members (ENACW$_{16}$ and SPMW$_7$) by the colder ones (ENACW$_{12}$ and IrSPMW) is in agreement with the observed cooling of the surface waters over the Reykjanes Ridge and east of 20°W in 2014 relative to 2002–2012 (Zunino et al., this issue).



The proportion of $SPMW_8$ generally increases in 2014 relative to 2002–2010 (Fig. 6c), replacing MW east of the Azores–Biscay Rise (Fig. 6f) and replacing $ENACW_{12}$ west of the Rise (Fig. 6b). The cooling and freshening at the level of MW and the surface cooling observed by Zunino et al. (this issue) supports the replacement of the warm and salty MW and $ENACW_{12}$ by the relatively cold and fresh $SPMW_8$.

In the first 1000 dbar around and over the Reykjanes Ridge, $SMPW_7$ markedly decreases in 2014 relative to 2002–2010 (Fig. 6d). In the surface layer, IrSPMW (Fig. 6a) replaces $SPWM_7$, while at greater depth is LSW (Fig. 6e) which replaces $SPMW_7$. These water mass redistributions agree with the observed cooling in these regions (Zunino et al., this issue), since IrSPMW and LSW are colder than $SPMW_7$. The positive surface anomalies of $SMPW_7$ east of 28ºW produce a decrease in $ENACW_{12}$ (Fig. 6b) and $SAIW_6$ (Fig. 6g). The replacement of $ENACW_{12}$ by $SMPW_7$ is also related to surface cooling; while the

interchange between $SAIW_6$ and $SMPW_7$ is explained by the relatively warm and salty anticyclonic eddy found at the northern limit of the NAC (Zunino et al., this issue; their Fig. 5 and 7).

The anomalies of the distribution of LSW for 2014 with respect to 2002–2010 in the Irminger Sea and the Iceland Basin are generally positive above 1000 dbar and negative below 1000 dbar (Fig. 6e). This general pattern of LSW anomalies is coincident with the pattern of the oxygen anomalies between 2002–2012 and 2014 (Zunino et al., this issue; their Fig. 7). At

the central western part of the Irminger Sea, positive LSW anomalies coincide with negative anomalies of IrSPMW (Fig. 6a), while on the eastern Irminger Sea and over the Reykjanes Ridge positive LSW anomalies coincide with negative anomalies of $SPMW_7$ (Fig. 6d). The increase in the relatively cold LSW at expenses of the relatively warm SPMWs is in agreement with the observed cooling above 1000 dbar in the Iceland Basin and Irminger Sea with respect to 2002–2012 (Zunino et al., this issue; their Fig. 7a). The negative anomalies of LSW between 1000 and 2000 dbar coincide with positive anomalies of $SPWM_7$

(Fig. 6d), while between 2000 dbar and the seafloor, ISOW replaces LSW (Fig. 6b). The negative anomalies of LSW coincide with areas that present an increase in salinity with respect to 2002–2012 (Zunino et al., this issue; their Fig. 7). The redistribution between the relatively fresh LSW and the relatively saline $SPWM_7$ and ISOW represents the progressive salinization that classical LSW (our SWT) has been experiencing since its last formation event in the late 1990s (Sarafanov et al., 2012; Yashayaev and Loder, 2016). García-Ibáñez et al. (2015) also reported how the progressive salinization of LSW

since the late 1990s produced a progressive decrease in LSW and an increase in ISOW.

The changes in the distribution of IrSPMW in 2014 with respect to 2002–2010 are highly variable (Fig. 6a). The positive IrSPMW anomalies observed in the surface Irminger Sea coincide with negative anomalies of $SPMW_7$, redistribution attributed to surface cooling. While the increase in IrSPMW within the EGIC and the central Irminger Sea corresponds to a decrease in LSW (Fig. 6e), redistribution most likely caused by the narrowing of the Irminger Gyre in 2014 with respect to 2002–2012

(Zunino et al., this issue).

The EGIC presents an increase in IrSPMW (Fig. 6a) and a decrease in LSW (Fig. 6e) in 2014 with respect to 2002–2010. This water mass redistribution could be linked to the recent deep convection observed in the Irminger Sea (Yashayaev and Loder, 2016), resulting in the formation of IrSPMW in the NW of the Irminger Sea (Brambilla and Talley, 2008), which would be



transported by the EGIC towards the Labrador Sea. The EGIC also presents an increase in PIW in 2014 with respect to 2002–2010 (Fig. 6f), which would indicate a volume increase of Arctic waters.

The proportion of ISOW clearly increases in 2014 with respect to 2002–2010 (Fig. 6b). West of 22ºW (i.e., within the Subpolar Gyre), the increase in ISOW compensates the decrease in LSW (Fig. 6e), which reflects the salinization of LSW since its last

formation event (Sarafanov et al., 2012; Yashayaev and Loder, 2016). East of 22ºW, the increase in ISOW compensates the decrease in $NEADW_L$ linked to a decrease in silicic acid in the range 20–35 µmol $kg^{-1}$ (i.e., in the mixing zone between ISOW con la NEADW) in 2014 compared to 2002–2010 (Fig. S2). The uniform increase in ISOW in 2014 with respect to 2002–2010 is coincident with the observed salinization of the deep-bottom waters of the section (Zunino et al., this issue; their Fig. 7), and could indicate the persistent increase in the entrainment of SPMW into ISOW (Yashayaev and Dickson, 2008; Sarafanov et

al., 2010), which would transmit the salinization of SPMW (e.g., Flatau et al., 2003; Häkkinen and Rhines, 2004; Böning et al., 2006; Thierry et al., 2008) to the deep-ocean.

The proportion of DSOW increases in the bottom areas of the central Irminger Sea and decreases over the Greenland shelf with respect to 2002–2010 (Fig. 6g). The increase in DSOW coincides with a decrease in LSW (Fig. 6e), which is also coincident with negative anomalies of temperature and salinity and positive anomalies of oxygen observed by Zunino et al.

(this issue; their Fig 7), possibly caused by a deep intrusion of the DWBC to the centre Irminger Sea (Zunino et al., this issue; their Fig. 3). The decrease in DSOW over the Greenland shelf is compensated by an increase in ISOW (Fig. 6b), which coincides with positive anomalies of salinity and a negative anomalies of oxygen (Zunino et al. this issue; their Fig 7). The contrasting changes in DSOW corroborates the variability in the composition of DSOW (e.g., Macrander et al., 2005; Tanhua et al., 2008; Falina et al., 2012; van Aken and de Jong, 2012).

**4.2 Differences in the water mass volume transport between OVIDE 2014 and the average OVIDE 2002–2010**

In this section, we describe and discuss the observed changes in the water mass volume transport across the OVIDE line (Portugal to Greenland) between 2014 and 2002–2010 (Fig. 5).

The observed net volume transport for Central Waters ($ENACW_{16}$, $ENACW_{12}$ and $SPMW_8$) across the 2014 OVIDE line (13.8 ± 1.4 Sv; Fig. 5) is in agreement with the average value of 11.6 ± 1.4 Sv for 2002–2010 (García-Ibáñez et al., 2015) but lower

than the 19.6 ± 1.7 Sv for 1997 reported as the net transport of the NAC by Lherminier et al. (2007). The higher volume transport of Central Waters in 1997 is linked to the higher AMOC intensity reported for that year (23.3 ± 1.2 Sv; Lherminier et al. 2007) with respect to the AMOC intensity reported for 2014 (18.7 ± 3.0 Sv; Zunino et al., this issue).

The slight increase in the net volume transport of $SPMW_7$, from the average value of 2.6 ± 1.5 Sv for 2002–2010 to 3.2 ± 2.1 Sv for OVIDE 2014 (Fig. 5), is linked to the decrease in $SPMW_7$ over the eastern flank of the Reykjanes Ridge with respect

to 2002–2010 (Fig. 6d), where waters are flowing southwards.

The observed net volume transport of 1.4 ± 0.6 Sv for SAIW ($SAIW_6$ and $SAIW_4$) across the OVIDE 2014 line (Fig. 6) is in agreement with the average value of 2.2 ± 0.4 Sv for 2002–2010 (García-Ibáñez et al., 2015), but is lower than the 2.9 Sv



reported by Álvarez et al. (2004) for 1997. The higher SAIW transport in 1997 compared to 2014 and 2002–2010 is related to the higher proportions of SAIW in 1997 and to the higher AMOC intensity in the year.

The transport of MW across the OVIDE line is highly variable, changing from 1.7 Sv in 1997 (Álvarez et al., 2004), to average value of $0.2 \pm 0.4$ Sv for 2002–2010 (García-Ibáñez et al., 2015) and $0.7 \pm 0.6$ Sv in 2014 (Fig. 6). Meddy activity may explain

the observed changes in the net transport of MW across the OVIDE line (Arhan and King, 1995; Mazé et al., 1997).

The net volume transport of IrSPMW for 2014 (-10.0 $\pm$ 1.2 Sv; Fig. 5) slightly increases from the average value of -8.9 $\pm$ 0.9 Sv for 2002–2010 (García-Ibáñez et al., 2015) due to the increase in IrSPMW in the EGIC (Fig. 1) with respect to 2002–2010 (Fig. 6a).

The net volume transport of LSW for OVIDE 2014 ($0.8 \pm 1.7$ Sv; Fig. 5) is in agreement with the $2 \pm 1$ Sv reported for OVIDE

2002 by Lherminier et al. (2007). However, García-Ibáñez et al. (2015) reported a net southward transport of LSW of -1.0 $\pm$ 1.8 Sv for OVIDE 2002–2010, very likely due to the presence of LSW in the location of the EGIC (Fig. 1), absent in 2014 (Fig. 6e).

The net volume transport of the shallow core of PIW across OVIDE 2014 (-2.2 $\pm$ 0.2 Sv; Fig. 5) is in agreement with previous estimates of around -2 Sv entering from the Arctic Ocean (barely -2 Sv reported by Pickart et al. (2005), and an average

transport of -2.4 $\pm$ 0.3 Sv reported by Falina et al. (2012) for 2002–2004). However, the net volume transport of PIW in 2014 is significantly higher than the average value of -1.4 $\pm$ 0.1 Sv for OVIDE 2002–2010 (García-Ibáñez et al., 2015). The increase in the contribution of PIW to the EGIC (Fig. 6f) explains the observed increase in the transport of PIW in 2014.

The net volume transport of DSOW across OVIDE 2014 (-2.2 $\pm$ 0.4 Sv; Fig. 5) is in good agreement with the estimates of Ross (1984) (from -2 to -3 Sv), Eden and Willebrand (2001) (-2.5 Sv), Lherminier et al. (2010) (-2 Sv for the OVIDE sections

of 2002 and 2004), and García-Ibáñez et al. (2015) (-2.4 $\pm$ 0.3 Sv for OVIDE 2002–2010). However, our estimate is slightly lower than the -3 Sv widely recognized as the long-term average transport of DSOW (e.g., Dickson and Brown, 1994; Macrander et al. 2005; Jochumsen et al. 2012). Mismatches when assessing water mass volume transports through dilutions of ''pure'' SWTs (OMP-based) instead than by isopycnal transport are expected since more than one water mass can be found inside the isopycnal ranges bounding physical water mass definitions. The slight decrease in the net volume transport of DSOW

between 2014 and 2002–2010 is linked to the decrease of its contribution within the EGIC (Fig. 6g).

The net southward transport of ISOW for OVIDE 2014 (-4.8 $\pm$ 1.1 Sv; Fig. 5) is significantly higher than previous estimates of about -3 Sv (e.g., Saunders, 1996; van Aken and Becker, 1996; Lherminier et al., 2007; Sarafanov et al., 2012). The ISOW transport is also significantly higher than the average value of -2.8 $\pm$ 0.8 Sv for OVIDE 2002–2010 (García-Ibáñez et al., 2015). The increase in the contribution of ISOW in 2014 (Fig. 6b) related to the salinization of the bottom areas of OVIDE

2014 (Zunino et al., this issue) is the responsible of the higher than expected net volume transport of ISOW.

Finally, the NEADW$_L$ net volume transport across OVIDE 2014 ($0.3 \pm 1.7$ Sv; Fig. 5) is compatible with previous estimates of about 1 Sv (e.g., van Aken and Becker, 1996). The relative decrease in the net volume transport of NEADW$_L$ between 2014 and 2002–2010 ($0.6 \pm 1.2$ Sv) is linked to the decrease of its contribution in the Iberian Abyssal Plain (Fig. 6g).





The assessment of the relative contribution of each water mass to the AMOC across the OVIDE line allows identifying the water masses involved in the slight increase in the AMOC intensity from the average value of 16.2 ± 2.4 Sv for OVIDE 2002–2010 to 18.7 ± 3.0 Sv for OVIDE 2014 (Zunino et al., this issue). An increase in the AMOC intensity implies an increase in the net northward and southward transports of its upper and lower limbs, respectively. The increase in the AMOC intensity is

related to the increase in the northward transport of the Central Waters in its upper limb, and to the increase in the southward flow of IrSPMW and ISOW in its lower limb.

## 5 Conclusions

We described and discussed the distribution of water masses along the GEOVIDE cruise (GEOTRACES-GA01 section), crossing the subpolar North Atlantic Ocean and the Labrador Sea in summer 2014. We also provided the relative contribution

from each water mass to the transports across the OVIDE line (Portugal to Greenland) of the GEOTRACES-GA01 section by combining the eOMP results with the velocity field. The water mass structure along the GEOTRACES-GA01 section, obtained through an eOMP analysis, is consistent with generally accepted knowledge of the Subpolar North Atlantic circulation, with the exception of high presence of ISOW. Our estimates of water mass transports are in good agreement with previous studies and match the main features of the northern North Atlantic Circulation, with the exception of higher than expected transport

of ISOW linked to the high proportion of this water mass along the section.

We also assessed the change in the water mass distribution and transport of the 2014 OVIDE line comparing them with the average OVIDE 2002–2010. At the upper and intermediate water levels, the colder end-members of Central Waters and SPMW, and LSW replaced warmer water masses in 2014 with respect to 2002–2010, in agreement with the observed cooling of the surface and intermediate waters. Below 2000 dbar, ISOW presents greater proportions in 2014 than in 2002–2010,

increase related to the salinity increase observed since the beginning of the OVIDE programme (2002). The higher than expected proportion of ISOW in the Irminger Sea and Iceland Basin is linked to the salinization of SPMW transmitted to both ISOW through entrainment and LSW through lateral mixing. While for the West European Basin, the higher than expected proportion of ISOW is linked to the decrease in NEADW$_L$ associated with the observed decrease in silicic acid in the mixing space between ISOW and NEADW. We also observed an increase in IrSPMW within the EGIC in 2014 with respect to 2002–

2010, which supports the recent deep convection events in the Irminger Sea. This comparison of the water mass distribution for OVIDE 2014 and 2002–2010 highlights the utility of the OMP analysis in the identification of temporal variations in water mass distributions linked to circulation changes and water mass transformation.

The quantification of the volume transport for each water mass into the two limbs of the AMOC allowed us to identify the water masses implicated in the strengthening of the AMOC at the OVIDE line in 2014 in relation to the average 2002–2010.

The increase in the intensity of the AMOC upper limb is related to the increase in the northward transport of the Central Waters, which is partially compensated by the increase in the southward flow of IrSPMW and ISOW in the AMOC lower limb.



**Author Contributions**

All authors contributed extensively to the work presented in this paper. M.I.G.-I., F.F.P. and P.L. designed the research. M.I.G.-I., F.F.P., P.L., P.Z. and P.T. analysed the physical and chemical data. M.I.G.-I. and F.F.P. developed the code for processing the data. M.I.G.-I. wrote the manuscript and prepared all figures, with contributions from all co-authors.

**Acknowledgements**

We are grateful to the captains, staff and researchers who contributed to the acquisition and processing of hydrographic data. We are special grateful to Morgane Gallinari, Emilie Grosstefan, and Manon Le Goff who contributed to the analysis of nutrients; and to the technical team: Pierre Branellec, Floriane Desprez de Gésincourt, Michel Hamon, Catherine Kermabon, Philippe Le Bot, Stéphane Leizour, Olivier Ménage, Fabien Pérault and Emmanuel de Saint-Léger. For this work M.I. Garcia-
Ibáñez and F.F. Pérez were supported by the Spanish Ministry of Economy and Competitiveness through the BOCATS (CTM2013-41048-P) project co-funded by the Fondo Europeo de Desarrollo Regional 2014–2020 (FEDER). P. Lherminier was supported by IFREMER. P. Zunino was supported by CNRS and IFREMER, within the framework of the projects AtlantOS (H2020-633211) and GEOVIDE (ANR-13-BS06-0014-02). P. Tréguer was supported by the LABEX-Mer (French Government "Investissement d'Avenir" programme, ANR-10-LABX-19-01).

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





**Table 1:** Properties characterising the Source Water Types (SWTs, see footnote a) considered in this study with their corresponding standard deviations. The square of correlation coefficients ($r^2$) between the observed and estimated properties are also given, together with the Standard Deviation of the Residuals (SDR) and the SDR/ε ratios from the data below 400 dbar. The ε (standard deviation of the water sample properties) used to compute the SDR/ε ratios are listed in Table S1. The last column accounts for the uncertainties in the SWTs contributions expressed on a per one basis.

| | $\Theta$ (ºC) | S | $O_2^0$ ($\mu$mol kg$^{-1}$) | $Si(OH_4)^0$ ($\mu$mol kg$^{-1}$) | $NO_3^0$ ($\mu$mol kg$^{-1}$) | Uncertainty |
|---|---|---|---|---|---|---|
| $ENACW_{16}$ | $16.00 \pm 0.15$ | $36.20 \pm 0.02$ | $246 \pm 6$ | $1.99 \pm 0.12$ | $0.00 \pm 0.15$ | 0.005 |
| $ENACW_{12}$ | $12.30 \pm 0.18$ | $35.66 \pm 0.03$ | $251 \pm 8$ | $1.2 \pm 0.9$ | $8.0 \pm 1.1$ | 0.017 |
| $SPMW_8$ | $8.00 \pm 0.11$ | $35.23 \pm 0.02$ | $289 \pm 6$ | $2.9 \pm 1.9$ | $11.4 \pm 1.3$ | 0.027 |
| $SPMW_7$ | $7.07 \pm 0.07$ | $35.160 \pm 0.006$ | $280 \pm 8$ | $5.29 \pm 0.15$ | $12.83 \pm 0.15$ | 0.024 |
| IrSPMW | $5.00 \pm 0.17$ | $35.014 \pm 0.011$ | $310 \pm 9$ | $5.9 \pm 0.4$ | $14.2 \pm 0.4$ | 0.046 |
| LSW | $3.00 \pm 0.17$ | $34.87 \pm 0.02$ | $302 \pm 10$ | $8.4 \pm 0.7$ | $15.3 \pm 0.7$ | 0.039 |
| $SAIW_6$ | $6.0 \pm 0.2$ | $34.70 \pm 0.03$ | $297 \pm 9$ | $5.6 \pm 2.4$ | $13.4 \pm 1.2$ | 0.024 |
| $SAIW_4$ | $4.5 \pm 0.2$ | $34.80 \pm 0.03$ | $290 \pm 9$ | $1.2 \pm 2.4$ | $0.0 \pm 1.2$ | 0.010 |
| MW | $11.7 \pm 0.2$ | $36.500 \pm 0.011$ | $190 \pm 7$ | $6.33 \pm 0.15$ | $13.3 \pm 0.2$ | 0.008 |
| ISOW | $2.60 \pm 0.08$ | $34.980 \pm 0.003$ | $294 \pm 9$ | $12.6 \pm 0.9$ | $13.8 \pm 0.6$ | 0.047 |
| DSOW | $1.30 \pm 0.06$ | $34.905 \pm 0.006$ | $314 \pm 10$ | $7.0 \pm 0.5$ | $12.9 \pm 0.8$ | 0.023 |
| PIW | $0.0 \pm 0.2$ | $34.65 \pm 0.03$ | $320 \pm 11$ | $8.4 \pm 2.5$ | $12.9 \pm 1.2$ | 0.012 |
| $NEADW_U$ | $2.50 \pm 0.08$ | $34.940 \pm 0.007$ | $272 \pm 10$ | $29.7 \pm 0.6$ | $18.2 \pm 0.5$ | [b] |
| $NEADW_L$ | $1.98 \pm 0.03$ | $34.895 \pm 0.003$ | $252 \pm 10$ | $48.0 \pm 0.3$ | $21.9 \pm 0.5$ | 0.014 |
| $r^2$ | 0.9999 | 0.9986 | 0.9938 | 0.9988 | 0.9938 | |
| SDR | 0.009 | 0.004 | 2 | 0.4 | 0.2 | |
| SDR/ε | 0.9 | 0.4 | 1.8 | 1.3 | 1 | |

[a]$ENACW_{16}$ and $ENACW_{12}$ = Eastern North Atlantic Central Water of 16ºC and 12ºC, respectively; $SPMW_8$, $SPMW_7$ and IrSPMW = Subpolar Mode Water of 8ºC, 7ºC and of the Irminger Sea, respectively; LSW = Labrador Sea Water; $SAIW_6$ and $SAIW_4$ = Subarctic Intermediate Water of 6ºC and 4ºC, respectively; MW = Mediterranean Water; ISOW = Iceland–Scotland Overflow Water; DSOW = Denmark Strait Overflow Water; PIW = Polar Intermediate Water; and $NEADW_U$ and $NEADW_L$ = North East Atlantic Deep Water upper and lower, respectively.

[b]No uncertainty is given for $NEADW_U$ since it is was decomposed between MW, LSW, ISOW and $NEADW_L$ (see Sect. 2.3).





**Figure 1: Schematic diagram of the large-scale circulation in the subpolar North Atlantic adapted from Rhein et al. (2011) and Zunino et al. (this issue). Isobaths every 1000 dbar are represented by black contours. GEOVIDE (GEOTRACES-GA01 section) hydrographic stations are indicated by yellow dots. The main topographical features are labelled: Azores–Biscay Rise (ABR), Bight Fracture Zone (BFZ), Charlie–Gibbs Fracture Zone (CGFZ), Iberian Abyssal Plain (IAP) and Mid Atlantic Ridge (MAR). The main water masses and currents are also represented: Denmark Strait Overflow Water (DSOW), Deep Western Boundary Current (DWBC), East Greenland Irminger Current (EGIC), Iceland–Scotland Overflow Water (ISOW), Irminger Current (IC), Labrador Current (LC), Labrador Sea Water (LSW), Mediterranean Water (MW), North Atlantic Current (NAC), and North East Atlantic Deep Water lower (NEADW$_L$).**



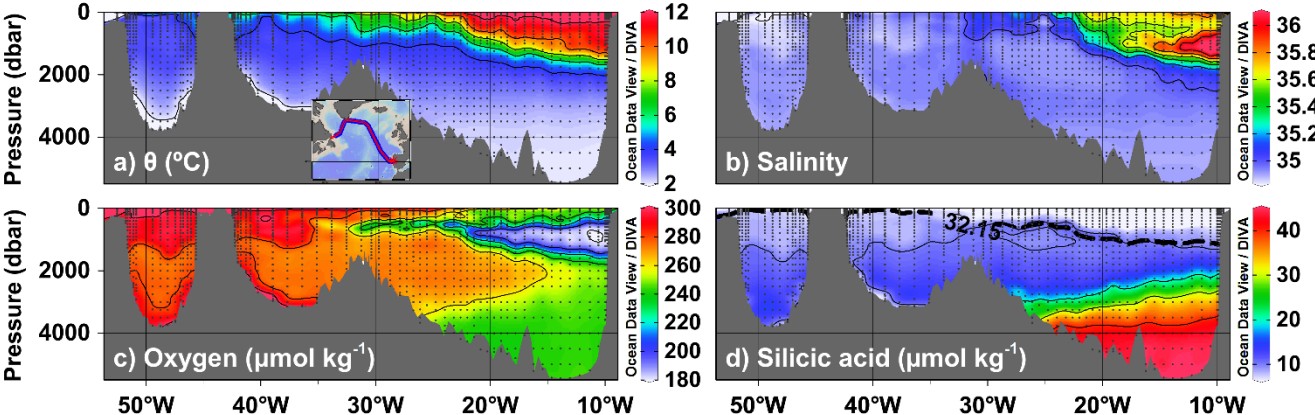

**Figure 2: (a) Potential temperature (θ, in ºC), (b) salinity, (c) oxygen (in µmol kg⁻¹), and (d) silicic acid (in µmol kg⁻¹) along 2014 GEOVIDE cruise (GEOTRACES-GA01 section, inset in subplot (a)), from Portugal (right) to Canada (left). Sample locations appear as black dots. The dashed horizontal black line in subplot (d) represents the isopycnal $\sigma_1 = 32.15$ kg m⁻³ (where $\sigma_1$ is potential density referenced to 1000 dbar), which marks the limit between the upper and lower limbs of the Atlantic Meridional Overturning Circulation (AMOC) at the GEOTRACES-GA01 section (Zunino et al., this issue).**





**Figure 3: (a) Potential temperature (θ)/salinity (S)-diagram including the source water types (SWTs; Table 1) and (b) zoomed for deep waters. The isopycnal σ₁ = 32.15 kg m⁻³ delimiting the upper and lower limbs of the AMOC at the GEOTRACES-GA01 section is also plotted. Colour coding represents the mixing groups (legend on (c)), i.e. subsets of SWTs susceptible of mixing together. (c) Distribution of the mixing groups along the section.**





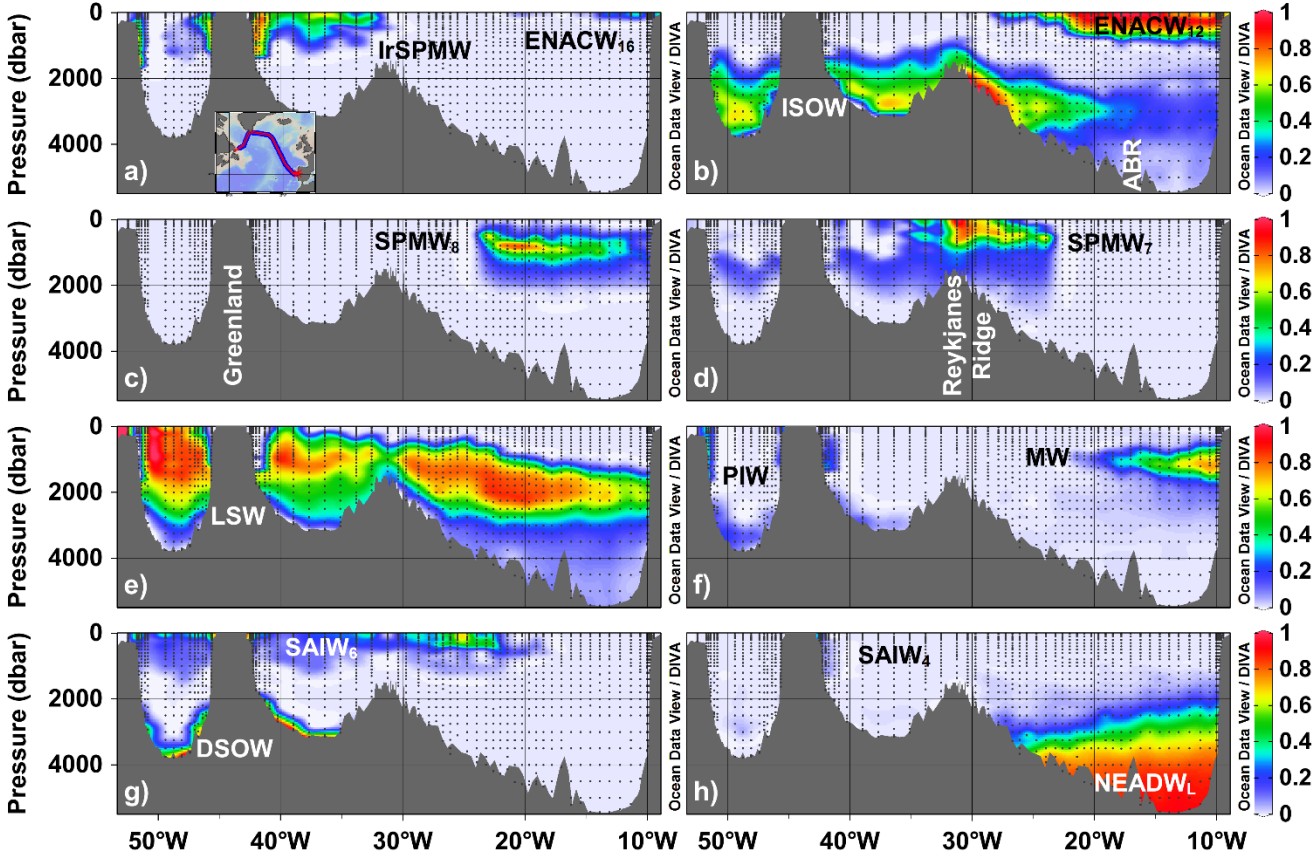

**Figure 4: Water mass distribution (on a per one basis) resulting from the eOMP analysis for the 2014 GEOVIDE cruise (GEOTRACES-GA01 section, inset in subplot (a)), from Portugal (right) to Canada (left). Sample locations appear as black dots. ABR refers to Azores–Biscay Rise. Confront Table 1 for water mass acronyms.**



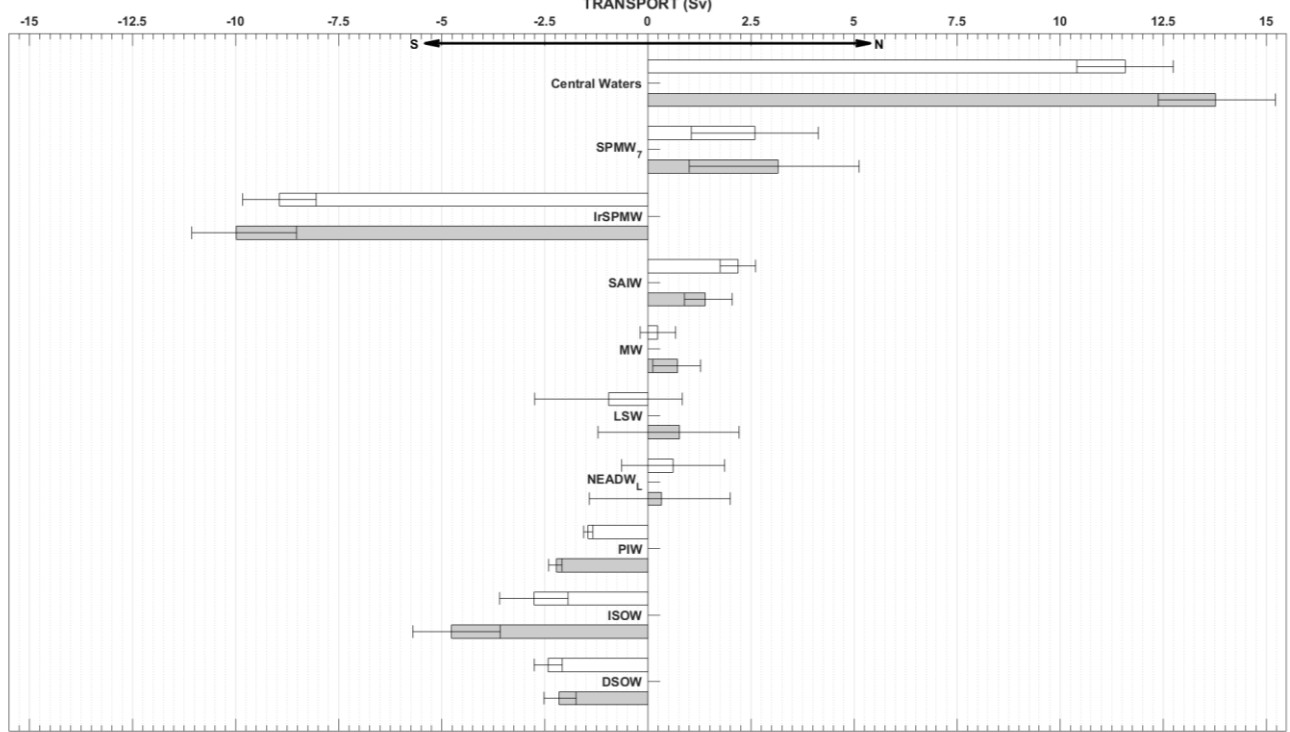

**Figure 5: Net water mass volume transports (in Sv; 1 Sv = $10^6$ m$^3$ s$^{-1}$) perpendicular to the OVIDE line for the 2002–2010 period (white bars; from García-Ibáñez et al. (2015)) and for 2014 (grey bars). Transports are positive (negative) northwards (southwards). Central Waters refers to the sum of ENACW$_{16}$, ENACW$_{12}$ and SPMW$_8$; and SAIW to the sum of SAIW$_6$ and SAIW$_4$ (confront Table 1 for water mass acronyms).**





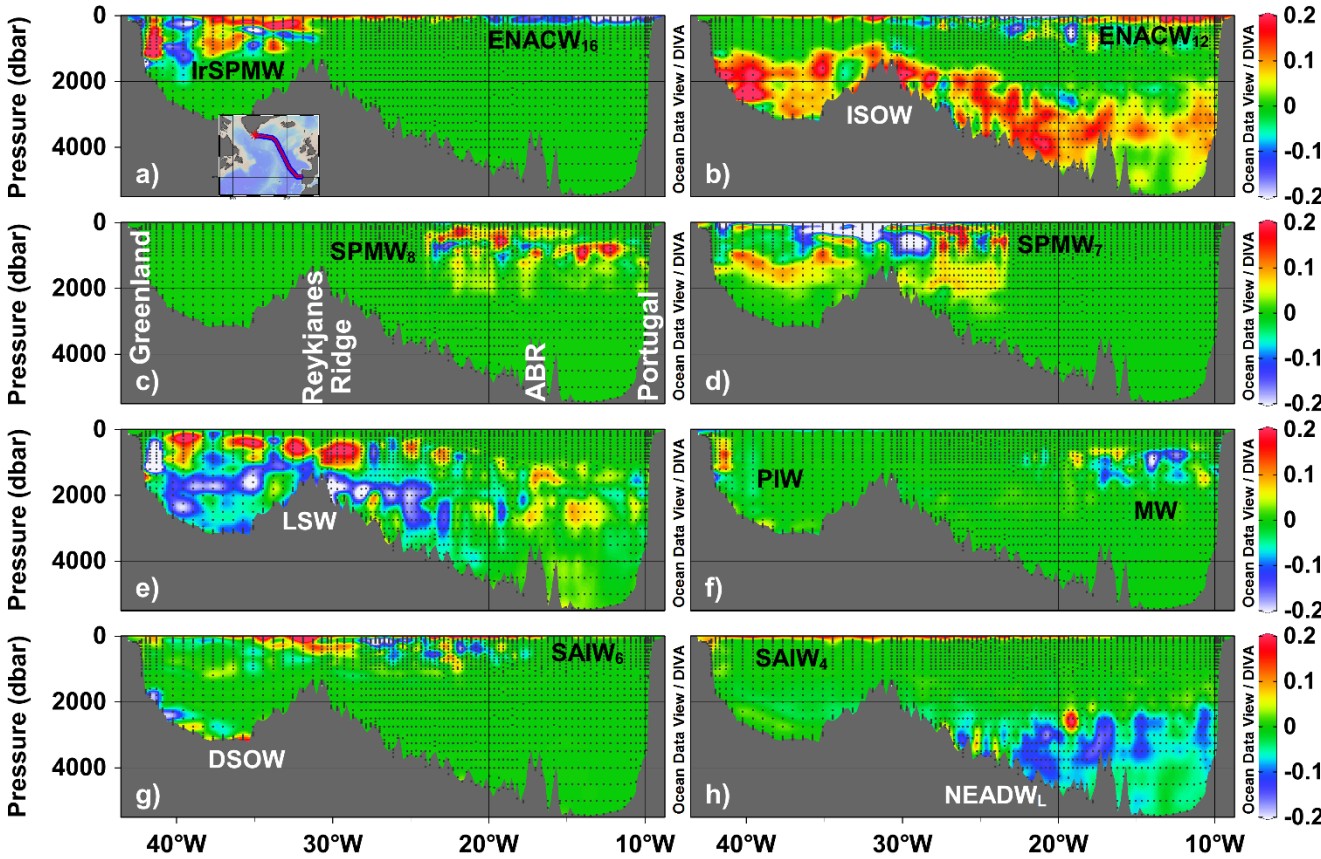

Figure 6: Water mass differences (on a per one basis) between the OVIDE line (inset in subplot (a)) in 2014 and the average of 2002–2010, from Portugal (right) to Greenland (left). Sample locations appear as black dots. ABR refers to Azores–Biscay Rise. Confront Table 1 for water mass acronyms.