# Peer review of "Water mass distributions and transports for the 2014 GEOVIDE cruise in the North Atlantic"

_Biogeosciences, 2017_

## Referee Comment (RC1) · Anonymous Referee #1 · 20 Nov 2017

Review: Water mass distribution and transports for the 2014 GEOVIDE cruise in the North Atlantic, Garcia-Ibanez et al.

***Summary of paper:***

Garcia-Ibanez et al. present data from a single section between Canada and Portugal in 2014 as part of the GEOTRACES programme. They use eOMP to decompose the water column into its constituent water masses and then combine these results with velocity estimates (Zunino et al., this issue) to estimate the transport of individual water masses (Greenland to Portugal only). Finally, the manuscript compares the 2014 results with the previously published 2002-2010 mean. The authors use the same methodology as Garcia-Ibanez et al., 2015, in part to enable easy comparison between the 2002-2010 mean and the 2014 results. However, this introduces some potential

problems in interpretation due to time-invariant SWT definitions. This paper is less sophisticated than Garcia-Ibanez et al., 2015 as it focusses on a single year's data and does a simple comparison with the long-term mean. However, I believe that the manuscript will be of use to those using and interpreting the GEOTRACES data. In particular I think the ability of eOMP to identify the individual water masses is a powerful tool.

***Main / Science comments:***

(1) The main problem I have with the manuscript is section 4.1, and the interpretation of an increase in the proportion of ISOW.

(i) The authors use time-invariant SWT definitions. However, it is known that the LSW definitions in particular vary temporally related to the intensity and depth of winter convection in the Labrador Sea. LSW has become warmer and more saline recently - as noted by the authors e.g. P10, L21-24. Surely an alternative explanation for the apparent increase in the proportion of ISOW is that it is an artefact of a salinification of the LSW source water whilst the eOMP uses a constant SWT? (And potentially also as a result of an increase in the salinity of Faroe Bank Channel bottom water: Hansen et al., 2016, Ocean Science, 12, doi: 10.5194/os-12-1205-2016)

(ii) I also disagree with the statement on P11, L8-10 that the observed salinization of the deep-bottom waters of the section supports the idea that more SPMW is entrained into ISOW. Firstly the eOMP results (Fig. 6) does not show increased SPMW in the ISOW region. Secondly Zunino et al., Fig 7 suggests to me that the salinification has occurred in the LSW rather than the ISOW.

(2) Similarly, I think that the ISOW transport discussed in Section 4.2 and the conclusion is higher as a result of the LSW (and ISOW?) salinification, rather than because of an increase in ISOW per se.

(3) I feel that the final paragraph of section 2.3 discussing the robustness of the eOMP

is important, but there are points that I don't understand:

(i) what standard deviation did you perturb the SWTs by? (i.e. what is the standard deviation in Table 1).

(ii) re. the uncertainties in the last column of Table 1 – is this the uncertainty introduced if just that SWT is perturbed? What does this tell us about total errors e.g. if more than one SWT is perturbed at the same time?

(iii) I find the last sentence in the paragraph about the correlation coefficients difficult to follow. Is it the same as in Garcia-Ibanez et al., 2015? If so the corresponding sentence in that paper is clearer: 'the model's ability to reproduce the measured values is given as the correlation coefficient (r2) between the measured (water samples) and the expected values for the SWTs properties (values of the properties of each water sample obtained by substituting Xi's in equation 3). The r2 values are higher than xxx indicating again the reliability of our method.'

(iv) Table S1. I don't understand how these standard deviations were generated? It says they are almost the same as the accuracies, but in section 2.1 the given accuracies are quite different to those in table S1. Maybe expand table heading?

***Minor Comments:***

(1) P7, L3-8: mention that very little SAIW4 is seen in the section, almost looks like it's not present from Fig. 4h

(2) P7, L18: also suggest referencing de Jong and de Steur, 2016, GRL, 43, doi10.1002/2016GL069596 for measurements in the Irminger Sea.

(3) Section 3.2: Why have you only calculated volume transports for the Greenland-Portugal part of the section?

(4) P8, L30: Can you say anything about how errors associated with the eOMP will contribute to the water mass volume transport errors? Or are the water mass volume

errors a lower bound estimate because no errors in the eOMP are taken into account?

(5) P9, L5: please could you add a few words / sentence describing how Zunino et al. defined the AMOC intensity?

(6) P10, L33: reference should really be de Jong and de Steur, 2016 rather than Yashayaev and Loder, 2016. de Jong and de Steur, 2016 present results from the Irminger Sea whereas Yashayaev and Loder, 2016 focus on the Labrador Sea.

(7) Section 4.2 Think need to mention that most changes in water mass volume transports between 2014 and 2002-2010 mean are within errors, with the exception of PIW and maybe just ISOW.

(8) P12, L22: please add more up-to-date reference for the DSOW transport estimates, Jochumsen et al., 2017, JGR Oceans, 122, doi:1 0.1002/2017JC012803.

***Comments figures:***

(1) Figure 3: please can you check the colour for group 10, as it seems to be different between a and c?

(2) Figure 3: there seems to be one dot on the Canadian Shelf that has been assigned to group 4 when it looks as if it maybe more appropriate to be assigned to group 1?

(3) Figure 4 caption: don't think need '(on a per one basis)'

(4) Figure 4 caption: 'Consult Table 1….' rather than 'Confront Table 1….'

(5) Figure 5 caption: 'Consult Table 1….' rather than 'Confront Table 1….'

(6) Figure 5 caption: Add sentence about what error bars are

(7) Figure 5: consider moving IrSPMW down, so have all water masses that contribute to upper limb of AMOC, and then all water masses that contribute to lower limb (?)

(8) Figure 6 caption: don't think need '(on a per one basis)'

(9) Figure 6 caption: 'Consult Table 1....' rather than 'Confront Table 1....'

(10) Figure 6: consider using different colour-scale e.g. one that has white around 0, warm colours for positive anomalies and cool colours for negative anomalies.

(11) Figure S1: please make axis lines thicker

(12) Figure S1: what are the units for plot a?

(13) Figure S2: consider using different colour-scale e.g. one that has white around 0, warm colours for positive anomalies and cool colours for negative anomalies.

***English / Typo suggestions:***

(1) P2, L7: insert word '....that enable us to trace back.....'

(2) P2, L15: insert word '..... as in the subpolar....'

(3) P2, L22: replace 'in' with 'of' '..... consisted of 78....'

(4) P2, L23-24: don't need to mention Short, Large, etc stations.

(5) P2, L25: mention what SBE43 is

(6) P2, L25: insert word '..... was used as a reference....'

(7) P2, L25: insert word '..... reference for the physical.....'

(8) P2, L27: replace 'in' with 'at' '.... performed at all.....'

(9) P2, L28: don't need CFA abbreviation as not used again in manuscript

(10) P2, L29: 'dividing' not 'diving'

(11) P2, L30: please give the accuracy for nutrients in $\mu$mol kg-1 rather than $\mu$M

(12) P3, L21: insert 'that' '.... NEADW that can be.....'

(13) P3, L33: don't use 'lcSPMW' anywhere else in the paper. Do you mean SPMW7?

(14) P4, L14: 'constraint' not 'constrain'

(15) P4, L22-25: don't think you need these lines. Mention it separating out biological and mixing components will be valuable when interpreting TEI distributions but then don't show in manuscript!

(16) P5, L6-8: re-order words '......Iceland Basin, with the $\theta$-S of SPMW8 being representative of that formed within the Iceland Basin....... and the $\theta$-S of SPMW7 to that found over the eastern....'

(17) P5, L18: 'crossed' not 'crossing'

(18) P5, L26: remove 'set'

(19) P5, L29: 'was' not 'were' '.... NEADWU was .....'

(20) P6, L6: change to '...allowed an assessment of.....'

(21) P6, L9: change word order '.....(Fig 4a.b), with ENACW12 being....'

(22) P6, L28: insert 'to' '.... NAC leads to the formation.....'

(23) P7, L4: change word order '..., with SAIW6 being the....'

(24) P7, L10: insert 's' '.... LSW concentrations reaching.....'

(25) P7, L11: insert 'the' '.... from the surface....'

(26) P7, L15: change word order '.... with the first 1000 dbar of the Irminger Sea being dominated.........'

(27) P7, L16: 'mixture' not 'mixing'

(28) P7, L18: insert 'water' '..... LSW-like water.....'

(29) P7, L23:24: change word order 'Some authors refer to the admixture.......... found over and around the Reykjanes Ridge as Icelandic Slope Water...........'

(30) P8, L22: 'hydrographic' rather than 'hydrological' ?

(31) P9, L5: insert words '. . .. intensity of the AMOC.. ..'

(32) P9, L24: insert 'our' 'Since our OMP analysis.. ...'

(33) P9, L24: remove 'the' '. . ... time invariant properties.. ...'

(34) P10, L6: add 's' and 'it' '. . .. greater depths it is LSW.. ...'

(35) P10, L17: add 'the' '. . .. at the expense.. ...'

(36) P10, L29: add 'with the' '(Fig 6e), with the redistribution.. ...'

(37) P11, L7: change 'con la'

(38) P11, L16: do the authors mean Greenland Slope rather than Greenland Shelf?

(39) P12, L12: insert 'an' '. . ... to an average.. ...'

(40) P13, L1: change to '. . ... allows identification of the water. . .. ...'

(41) P13, L20: insert 'with the' '. . .2002-2010, with the increase related.. ...'

(42) Abstract, P1, L19: insert 's' '. . ....colder end-members of the.. ...'

(43) Abstract, P1, L21: insert 'with the' '. . ... 2002-2010, with the increase. . ..'

(44) Abstract, P1, L25: 'identification of' not 'identifying'

---

## Referee Comment (RC2) · Anonymous Referee #2 · 27 Dec 2017

Good day,

The first reviewer has already provided a summary of the paper, so I will just go to straight to my points. However, everything else I say below solely reflects my opinion and view on the complex process of water mass formation and variability in the North Atlantic.

The issues the authors address in the paper are highly relevant and important for water mass analysis and prediction of their changes over time, and dissecting transformation and mixing of water mass is a big and nontrivial problem overall, so any novel solid approach and a study based on it would be much awaited here.

However, I cannot understand how a trans-Atlantic snapshot (not to mention that the

section does not end in St. John's, Newfoundland) and a simple model operating with only four members at once can be used to depict complex interaction and mixing of 14 water masses. I am not in position to judge the previously published paper of the same authors that is used as a basis for the current one, but if I had to review it, I would come with critical suggestions pretty much similar to those presented below.

Let me explain why I believe that a four-member approach does not work for this specific task:

(1) First of all - the case is not two-dimensional (2D distance along section vs depth). The water masses interact in over the entire subpolar North Atlantic. So, for example, any two waters appearing as neighbors on the OVIDE line may be separated by other waters elsewhere in the region. Therefore, the only way to solve this problem for the subpolar North Atlantic and its water masses is through solving a full system of equations where each end-member is carefully defined, and this creates another challenge.

(2) Now, a whole list of problems concerning the end members: a) The authors use end-member properties as they appear on a snapshot of an arbitrary section line (OVIDE or any) and not the properties of the studied water masses that these waters acquired at the times of their formation. Most critically here, both DSOW and ISOW should start from sub-zero temperatures. Both ISOW and DSOW are equally fresh the sills. However, ISOW gains its salt through mixing as it spread through the Iceland Basin. So taking the water that is already salty is not good for telling how it was formed from start – note that it has already been mixed with SPMW. Same is true about the other waters. (b) By no means, LSW remains undiluted between Labrador Sea and Iceland Basin. However, Figure 4 suggests 90% of original LSW in any other LSW all the way through the region. Well, the Labrador Sea is a very powerful engine, but can it pump so much water that stays unmixed for so far and so long? (c) The depth of LSW was not 2000 m in 2014, and there cannot 50% of LSW at 3000 m - at the depths where water is already as saline as ISOW modified through entrainment.

(d) Then, ISOW is fresher in Labrador Sea than in the Irminger Sea, because it is more diluted, but the corresponding fractions seem very much comparable in Figure 4. Does ISOW really reach 2000 m in the Labrador Sea adding about 50%? Or is it something else? How can we be so sure that another water mater contributing to the mid-depth exchanges and arriving from outside the Labrador Sea is not missed in this formulation? It must be something else rather than 50% of ISOW... (e) I totally agree that a more careful approach is needed for the two chemical variables used in the work. However, using a certain universal model for utilization may lead to overconsumption of oxygen at greater depth. I say this, because the oxygen section suggests weak biological utilization, whereas applying parameterizations used in biochemistry (I cannot expand further here, but any quick assessment would show a comparable result) would reduce dissolved oxygen more than what we see in the section. If we assume a strong bio-consumption, then how would we explain that dissolved oxygen closely follows salinity which in turn is not altered by living organisms?

So far I was talking about using static end members assuming the picture does not change with time. But there is another set of complications coming into play if we introduce temporal variability of water properties. Yes, the source waters change in time, but any static model assumes invariance of the source waters. How long does it take for LSW to cross the basin? Let's say N years? How would the authors introduce the temporal changes previously observed in the source or sources of LSW? Note that convection was not strong in 2010 and 2011, and that it was that water that had probably been seen in the Iceland Basin in 2014! LSW does change a lot in its source in 3-4 years. How would this knowledge be transpired into 3.00 and 34.87 with such narrow error bars? At the season of formation the waters are even more different. Oxygen saturation is probably >95%. Taking the transit time into consideration, the version of LSW seen on the OVIDE line in the southern Labrador Sea may not be directly related to that transferred to Iceland basin first through DWBC and then under NAC ... The properties of the original waters can be much greater than the error bars used through the work. I bring LSW only as an example but the same may true about

other waters brought into the equations.

Is it really true that DSOW has no LSW mixed into it? I find it strange because in the northern Imringer Sea DSOW is cascading down the slope entraining both NEADW (ex-ISOW) and LSW and warmer waters.

The Monte Carlo technique would only help if the errors were random respecting central tendency. I have no doubt that each of the linear 4-member solutions would converge even with larger seeded errors. However, the present case is subject to more systematic rather than random biases, raising a question like "How would each solution change if LSW was 0.3 warmer at time of formation?"

Saying that the task of unscrambling water mass composition in this highly dynamic and variable area is well worth pursuing, I, unfortunately, cannot agree that the presented method, data and results help much solving this task. There must be a solution, but based on a more extensive synthesis of three-dimensional (3D) data, on a proper definition of source waters and their changing properties, on accounting for multiple pathways, etc.

Concerning the transport part ... The water mass transport and transformation are two related problems. I don't think a simple geostrophy (note a coarse grid in the Irminger Sea and missing profiles in the western Labrador Sea - both are important for budgeting the fluxes) is sufficient for constraining the transports. Frankly, I would not even bring the transport part in the work discussing the contributions of source waters. I think the most important part for now is building a method adequate for the task and thoroughly investigating every aspect of interaction by taking into account a huge baggage of what is known and available to this date and developing something better than a static 2D approach for analysing a strongly time and space variant 3D dynamics and variability – essentially 4D.

To conclude my review I share my thinking of this problem as cookbook analogy – all we try to come up here with is a recipe. Think of real ingredients and not those appearing

someplace somewhere – if you use the latter, the results are not going to tell much about your true ingredients. On the other hand, by weighting the real properties of the waters with the sought and found fractions, one should come to a section plot similar to that observed.

Considering the amount of data, effort and work needed to address the issues I raised in my review, I recommend rejection. This only reason why it is not revision is that by redoing the paper the authors would come with a totally new method, sets of results and visions. Sorry, but I cannot see it any simpler than that.

---

## Referee Comment (RC3) · Anonymous Referee #3 · 4 Jan 2018

General comments

This paper presents a comprehensive description of all water types that cross the OVIDE line, which in itself is informative. If this has not been presented elsewhere in the literature, it is worth publishing. This work also illustrates some important water mass redistributions in 2014, with the more subarctic water masses generally replacing the more subtropical water masses. This is also interesting. But then we get too much information on water mass changes, based on a relatively unclear picture (Fig. 6). This part could be shortened/improved.

With the uncertainty associated with the water mass distribution, and the inherent noise in a single occupation of such an oceanographic section, I did not find all the transport details Section 4.2 interesting.

[Figure]

Much due to all the detail in the latter part of the manuscript, I still find this paper difficult to digest. The additional information on the eOMP and the water types helps to make sense of the paper contents. But still, as I came to the latter part of the Discussion, I really had to stay focused, in order to continue reading.

As it stands, it is mostly interesting to a rather limited inner circle of devoted oceanographers. It could, however, become a nice contribution is shortened and clarified. The writing generally reads well already.

Detailed comments

I am not familiar with the content in the much cited Zunino et al., which might limit my ability to interpret some of the findings in this manuscript.

The use of a subscript (e.g. SPMW7) for different types the otherwise well known water masses, is new to me. And this notation is not even introduced in this manuscript. I must admit that this detail hindered me in following this manuscript initially. This confusion might be caused by my lack of knowledge, but this will probably confuse other readers as well. Please give a better introduction to this, and improve the integration with the literature. E.g. how does the SPMW7 associated with the water mass descriptions given in other oceanographic papers?

This work seems to use different – and maybe lower - values for the nutrient concentrations in the SWTs, compared to some other studies. The authors e.g. use a silicic acid concentration of 6.33 $\mu$M to represent the MW, while (McGrath et al., 2012) use a silicate concentration of 10-11 $\mu$M for the same water mass.

a) Is silicic acid, $Si(OH4)0$, not the same as "silicate"? b) Why use both one and two decimals in Table 1? c) How sensitive is eOMP method to such different choices of the source water silicic acid concentration?

I am a bit confused by the definition and discussion of the 'Central Water'. On page 5, line 26 (p5,l26) this water mass is defined as ENACW16+ ENACW12, on p5,l29 is

stated that "The distribution of the Central Waters is associated with the NAC" and on (p3,l6) is stated that the Central Waters is transported with the NAC. How can it be defined by the eastern waters, and be transported with the NAC? Please clarify.

(p5,l6): "An important assumption of the methodology is that the physical and chemical characteristics of the SWTs are considered time invariant and..."

(p9,l31): "...the progressive salinization that classical LSW (our SWT) has been experiencing since its last formation event in the late 1990s..."

So the eOMP method seems to be importantly dependent on the assumption of time invariability of the SWTs, and it is clear that the SWT are not time invariable. At first glance, this appears as a contradiction. Please explain.

(p7,l32): "...measurements and by an overall mass balance of $1 \pm 3$ Sv northwards..." It is not clear what this means.

(p9,l13) and below: The abbreviation SMPW is often used. I assume this should be SPMW. Section 4.1. The discussion on the water mass changes between the average 2002-2012 state, and 2014 is difficult to follow. This is partly because Fig. 6 could be improved (see comments below), and partly because the patterns are not always clear. Maybe guide the reader better to the mentioned changes (e.g. specify depths levels).

(p9,l27): "The negative anomalies of LSW between 1000 and 2000 dbar coincide with positive anomalies of SPWM7... "

It makes sense that the cooling after 2014 was associated with a replacement of the relatively warm SPWM7 with the colder LSW. But it seems counterintuitive that the opposite occurred below 1000 dbar. Was LSW really replaced by the warmer SPMW at these deeper levels? Please explain. One result of this paper is an unexpectedly high presence of ISOW. It is known that the eOMP is sensitive to the assumption of time invariability of the SWTs, and it is clear that the ISOW SWT became more saline after 2002. Could the unexpectedly high presence of ISOW partly be a result of this

uncertainty?

Figures

Figures 1-5 are all ok.

The NAC in Fig. 1 is located farther south than where we usually see it (in the literature). Is this because the authors suggest that the NAC is actually located this far south?

Fig. 2d. The silicic acid values span 0-40 $\mu$M, probably in order to get the highest values near the seafloor in the eastern part represented as well. But most of the observed silicate variability is seen in the range 2-12 $\mu$M, and the figure has a low resolution in this range. Maybe consider using a non-linear color code?

Figure 6

The message in this figure is not clear.

a) Maybe use different software. Although ODV is well suited to scan oceanographic data, it might not be the right choice for making publishable figures. If you still want to use ODV, remove the redundant references to this software.

b) Use the y-axis label, "Pressure (dbar)" only once. The same goes for the other ODV-based figures (Figs. 2 and 4).

c) What does "(on a per one basis)" really mean?

d) Maybe add something a la the text in (p9,l4) "Positive (negative) anomalies in the proportion of a water mass imply a gain (loss) in 2014 compared to 2002–2010." to the caption for Fig. 6.

e) Since the patterns in this figure are quite noisy, one can doubt the usefulness of this figure. The uncertainty about the parameter shown, and the definition of the water types with that subscript (e.g. ENACW16, see comment above), it becomes difficult to

follow the discussion related to this figure.

Suggestions: a) Show and discuss only the clearest signals (fewer panels in the figure). Patterns with blue and red blobs might too much associated with the inherent variability, which could strongly impact a single transect along the OVIDE line. Or b), improve the figure and the explanation of its content, and integrate the discussion with this figure in a clearer way.

References

McGrath, T., Nolan, G., McGovern, E., 2012. Chemical characteristics of water masses in the Rockall Trough. Deep-Sea Research Part I-Oceanographic Research Papers 61, 57-73.

---

## Author Comment (AC1) · 27 Feb 2018

*We thank referee #1 for the helpful comments. We specially thank the referee for the thorough English correction. We have addressed the referee's concerns as explained below.*

***Main / Science comments:***

(1) The main problem I have with the manuscript is section 4.1, and the interpretation of an increase in the proportion of ISOW.

(i) The authors use time-invariant SWT definitions. However, it is known that the LSW definitions in particular vary temporally related to the intensity and depth of winter convection in the Labrador Sea. LSW has become warmer and more saline recently – as noted by the authors e.g. P10, L21-24. Surely an alternative explanation for the apparent increase in the proportion of ISOW is that it is an artefact of a salinification of the LSW source water whilst the eOMP uses a constant SWT? (And potentially also as a result of an increase in the salinity of Faroe Bank Channel bottom water: Hansen et al., 2016, Ocean Science, 12, doi: 10.5194/os-12-1205-2016)

*We have performed a new OMP run where we slightly modified the temperature and salinity (TS) properties for LSW and ISOW to match those found in the most recent period and we revised the standard deviations of the properties that define the SWTs taking into account the temporal variability. We have used the results of this new OMP run as the final results of the manuscript. Even using the TS properties for LSW and ISOW closest to those observed in recent years, we obtained proportions of ISOW higher than the mean values reported in the literature. Therefore, we are confident that the higher than expected concentrations of ISOW is a real feature, which is consistent with the increase in the volume transport of ISOW observed in the OSNAP array (Johns et al., 2017; Zou et al., 2017). We have added this result in the manuscript:* "The uniform increase in ISOW is consistent with the increase in volume transport of ISOW observed in the OSNAP array (Johns et al., 2017; Zou et al., 2017)".

(ii) I also disagree with the statement on P11, L8-10 that the observed salinization of the deep-bottom waters of the section supports the idea that more SPMW is entrained into ISOW. Firstly the eOMP results (Fig. 6) does not show increased SPMW in the ISOW region. Secondly Zunino et al., Fig 7 suggests to me that the salinification has occurred in the LSW rather than the ISOW.

*We agree that the statement was not formulated correctly. Our OMP setting does not allow us to disentangle the causes of the changes in the properties of ISOW. As indicated in the answer to the previous comment, we have performed a new OMP run using new TS properties for ISOW and LSW. The TS properties of ISOW have been changed according to the observed warming and salinization in ISOW after crossing the Faroe Bank Channel (Hansen et al., 2016). This warming and salinization were due to a change in the properties in both the Nordic Seas waters and in the Atlantic waters entrained in the overflow (Hansen et al., 2016). Regarding the salinization reported by Zunino et al. (2017), their figure 7 shows a salinity increase in LSW and in deeper levels corresponding to ISOW. We have rewritten the entire section and the paragraph corresponding to ISOW now reads as:* "Below $\sigma_3 = 41.25$ kg m$^{-3}$, the decrease in the contribution of LSW in 2014 with respect to 2002–2010 is balanced by an increase in ISOW (Fig. 6b). This water mass redistribution responds both to the salinization of LSW (e.g., Yashayaev and

*Loder, 2017), and to the lower density of LSW formed in recent years that occupies shallower positions in the water column. García-Ibáñez et al. (2015) also reported how the progressive salinization of LSW since the late 1990s resulted in a progressive decrease in LSW and increase in ISOW. East of 22ºW, the increase in ISOW compensates the decrease in $NEADW_L$, which is linked to a decrease in silicic acid in the range of 20–35 µmol $kg^{-1}$ (i.e., in the mixing zone between ISOW and NEADW) in 2014 compared to 2002–2010 (Fig. S2). The uniform increase in ISOW is consistent with the volume transport increase in ISOW observed in the OSNAP array (Johns et al., 2017; Zou et al., 2017)".*

(2) Similarly, I think that the ISOW transport discussed in Section 4.2 and the conclusion is higher as a result of the LSW (and ISOW?) salinification, rather than because of an increase in ISOW per se.

*See answer to comment 1(i).*

(3) I feel that the final paragraph of section 2.3 discussing the robustness of the eOMP is important, but there are points that I don't understand:

(i) what standard deviation did you perturb the SWTs by? (i.e. what is the standard deviation in Table 1).

*We have added the information in the supplementary material and we have referenced it in a footnote added in Table 1.*

*"Text S1*
*The standard deviations (STD) of the potential temperature and salinity that define the source water types (SWTs) were taken from the literature. For Central Waters and SPMWs, the STDs were set as ± 0.6ºC for temperature and ± 0.06 for salinity, according to the thermohaline variability reported by Robson et al. (2016) for the first 700 m of the water column of the subpolar gyre. For LSW, the STDs were set as ± 0.4ºC for temperature and ± 0.01 for salinity, to include both the thermohaline properties used in García-Ibáñez et al. (2015) and those used in this work. For SAIW, the STDs were set as ± 0.5ºC for temperature and ± 0.03 for salinity, based on the variability of the thermohaline of its source waters, i.e., Central Waters and LSW (Iselin, 1936; Arhan, 1990; Read, 2000). For MW, the STDs were set as ± 0.2ºC for temperature and ± 0.07 for salinity, according to the work of Carracedo et al. (2016). For ISOW, the STDs were set as ± 0.1ºC for temperature and ± 0.02 for salinity, to include both the thermohaline properties used in García-Ibáñez et al. (2015) and those used in this work. For DSOW, the STDs were set as ± 0.16ºC for temperature and ± 0.008 for salinity, according to the work of Jocchumsen et al. (2012). For PIW, the STDs were set as ± 0.2ºC for temperature and ± 0.03 for salinity, according to the work of Falina et al. (2012). For $NEADW_L$, the STDs were set as ± 0.03ºC for temperature and ± 0.003 for salinity, according to the work of García-Ibáñez et al. (2015). For $NEADW_U$, the STDs for potential temperature and salinity were calculated using the STDs of its components: MW, LSW, ISOW and $NEADW_L$ (Sect. 2.3 of the main text).*
*For oxygen, the STDs were set equal to 3% of the saturation value (Najjar and Keeling, 2000; Ito et al., 2004), whereas for nutrients they were obtained by one of the following methods:*
*a)        For to LSW, ISOW and $NEADW_L$, the STDs for the nutrients was calculated using the STDs in the water samples with more than 95% of those SWTs, following Karstensen and Tomczak (1998). This method was used when the number of water samples for a SWT was greater than 50.*
*b)        For the Central Waters, DSOW and SPMW, which are defined by more than one SWT (multi-SWTs), the multi-SWT contributions were obtained by adding the contributions of their respective components. Then, water samples with proportions of the multi-SWT greater than 95% were selected. The property values of each component of the multi-SWT were subtracted from the values of the water samples and linear regressions were performed between potential temperature and nutrients. The STDs of the multi-SWT nutrients were taken equal to the error of the intercept. We used the STDs of the properties of the multi-SWTs to each of their components.*
*c)        A modification of the methodology (b) was applied to MW, where samples with proportions greater than 75% were selected to perform the linear regressions.*

*The STDs of the nutrients of SAIW were assigned equal to those of the Central Waters, because not enough water samples presented proportions greater than 95%. The STDs of the nutrients of NEADW$_U$ were calculated using the STDs of its components: MW, LSW, ISOW and NEADW$_L$ (Sect. 2.3 of the main text).*

*References:*

*Carracedo, L. I., Pardo, P. C., Flecha, S., and Pérez, F. F.: On the Mediterranean Water Composition, J. Phys. Oceanogr., 46, 1339–1358, https://doi.org/10.1175/JPO-D-15-0095.1, 2016.*

*Ito, T., Follows, M. J., and Boyle, E. A.: Is AOU a good measure of respiration in the oceans?, Geophysical Research Letters, 31, L17305, doi:10.1029/2004GL020900, 2004.*

*Jochumsen, K., Quadfasel, D., Valdimarsson, H., and Jónsson, S.: Variability of the Denmark Strait overflow: moored time series from 1996–2011, Journal of Geophysical Research, 117, C12003, doi:10.1029/2012JC008244, 2012.*

*Najjar, R.G., and Keeling, R.F.: Mean annual cycle of the air-sea oxygen flux: a global view, Global Biogeochemical Cycles, 14 (2), 573–584, doi:10.1029/1999GB900086, 2000".*

(ii) re. the uncertainties in the last column of Table 1 – is this the uncertainty introduced if just that SWT is perturbed? What does this tell us about total errors e.g. if more than one SWT is perturbed at the same time?

*The uncertainty shown in the last column of Table 1 results from perturbing the properties of all the SWTs and of all the water samples at the same time. During the perturbation process, the values of all the properties defining the SWTs and the values of all the samples are modified, and the OMP is solved for each perturbed system, obtaining the proportions of each SWT. The process is repeated 100 times, with the average of the 100 OMP solutions being the final result and the average standard deviations the uncertainty shown in Table 1. We have changed the text to make it clearer: "the properties **of both** each SWT and **each** water sample were perturbed". This allows a joint assessment of the sensitivity of the OMP analysis to both measurement errors and variations in the physical and chemical properties of the SWTs, as indicated in the manuscript.*

(iii) I find the last sentence in the paragraph about the correlation coefficients difficult to follow. Is it the same as in Garcia-Ibanez et al., 2015? If so the corresponding sentence in that paper is clearer: 'the model's ability to reproduce the measured values is given as the correlation coefficient (r2) between the measured (water samples) and the expected values for the SWTs properties (values of the properties of each water sample obtained by substituting Xi's in equation 3). The r2 values are higher than xxx indicating again the reliability of our method.'

*Thank you for your comment. As you mention, it refers to the same content as in García-Ibáñez et al. (2015). We have rewritten it accordingly: "We tested the robustness of the methodology through a Monte-Carlo simulation (Tanhua et al., 2005), where the physical and chemical properties of both each SWT and each water sample were randomly perturbed within the standard deviation of each parameter (see Text S1 and Table S1). This allowed an assessment of the sensitivity of the eOMP analysis to potential measurement errors and temporal variations in the physical and chemical properties that define the SWTs (Leffanue and Tomczak, 2004). A hundred Monte-Carlo simulations were performed and the eOMP equation system was solved for each of them. The average standard deviation of the Xis (last column in Table 1) is lower than 12%, which indicates that the methodology is robust. Additionally, our eOMP analysis is consistent since its residuals (r in Eq. 3) lack a tendency with depth (Fig. S1), with the standard deviations of the residuals being slightly higher than the measurement errors (Table 1). Besides, the ability of our eOMP analysis to reproduce the measured values is given as the correlation coefficient ($R^2$, Table 1) between the measured values*

*(water samples) and the expected values for the SWT mixing (values of the properties of each water sample obtained by substituting Xis in Eq. (3)). The $R^2$ values are higher than 0.993, which again indicates the reliability of our eOMP analysis".*

(iv) Table S1. I don't understand how these standard deviations were generated? It says they are almost the same as the accuracies, but in section 2.1 the given accuracies are quite different to those in table S1. Maybe expand table heading?

*The standard deviations of the properties of the water samples were taken close to the accuracy of each property. The difference between the accuracy of the measurement ($\theta$ 0.001, S 0.002, Si(OH)$_4$ 0.1, NO$_3$ 0.1, and O$_2$ 2) and the standard deviation ($\theta$ 0.01, S 0.01, Si(OH)$_4^0$ 0.3, NO$_3^0$ 0.2, and O$_2^0$ 1) is based on the fact that these standard deviations are used in the weighting process of the OMP. The OMP equations are weighted according to the accuracy of the property and/or to the variability in the region of study (Leffaune and Tomczak, 2004). Therefore, the standard deviations listed in Table S1 represent both the accuracy of the measurements and the variability in the study region. However, we have revised the standard deviation of temperature and salinity, to be in agreement with the general requirements of global datasets such as Glodapv2 (Olsen et al., 2016). We have changed the standard deviation of temperature and salinity to 0.005 and 0.002, respectively. We have changed the heading of Table S1 accordingly: "Standard deviations of the properties of the water samples ($\varepsilon$ in Table 1) were obtained by considering   **both** the accuracy of each  **measurement and the variability in the study region**".*

***Minor Comments:***

(1) P7, L3-8: mention that very little SAIW4 is seen in the section, almost looks like it's not present from Fig. 4h

*Thank you for your comment. We have added '...with SAIW$_6$ being the main end-member and SAIW$_4$ only found over the Greenland Slope with less than 35% of contribution (average of 11 ± 7 %; n=55)' (P7, L4 in the previous version of the manuscript).*

(2) P7, L18: also suggest referencing de Jong and de Steur, 2016, GRL, 43, doi10.1002/2016GL069596 for measurements in the Irminger Sea.

*Thank you for your suggestion. We have added the citation.*

(3) Section 3.2: Why have you only calculated volume transports for the Greenland-Portugal part of the section?

*We have been able to determine that the computation of the transports through the Greenland-Portugal transect of GEOVIDE was robust despite the subsampling of certain regions. A subsampling the previous OVIDE transects was performed in a similar way and it was verified that even with this subsampling the transports of the major currents were estimated correctly (see Zunino et al., 2017). This was not possible for the transect in the Labrador Sea for which we had no reference. Therefore, the velocity field for the Labrador Sea has not yet been solved.*

(4) P8, L30: Can you say anything about how errors associated with the eOMP will contribute to the water mass volume transport errors? Or are the water mass volume errors a lower bound estimate because no errors in the eOMP are taken into account?

*We have calculated the errors associated with the eOMP (the uncertainties listed in the last column of Table 1) in the water mass transports using a Monte-Carlo simulation. Then, we have propagated both errors (those associated with the uncertainty in the velocity field and those associated with the uncertainty of the water mass proportions) to calculate the uncertainty of the water mass transports. We have changed the sentence to take this change into account: "Errors were computed by  propagating both the uncertainty of  the $X_i$s (listed in Table 1) and the uncertainty of the velocity field".*

(5) P9, L5: please could you add a few words / sentence describing how Zunino et al. defined the AMOC intensity?

*We have added the following: "The AMOC intensity is defined as the maximum of the surface-to-bottom integrated stream function computed in density coordinates (Zunino et al., 2017)".*

(6) P10, L33: reference should really be de Jong and de Steur, 2016 rather than Yashayaev and Loder, 2016. de Jong and de Steur, 2016 present results from the Irminger Sea whereas Yashayaev and Loder, 2016 focus on the Labrador Sea.

*Thank you for suggesting this citation. We have performed the change.*

(7) Section 4.2 Think need to mention that most changes in water mass volume transports between 2014 and 2002-2010 mean are within errors, with the exception of PIW and maybe just ISOW.

*We have added the following text at the end of the first paragraph: "Most of the changes in the net water mass volume transports between 2014 and the 2002–2010 mean are within the errors and, therefore, are not significant, with the exception of SAIW, PIW and ISOW, which are further discussed".*

(8) P12, L22: please add more up-to-date reference for the DSOW transport estimates, Jochumsen et al., 2017, JGR Oceans, 122, doi:10.1002/2017JC012803.

*Thank you for suggesting this citation. We have performed the change.*

***Comments figures:***

(1) Figure 3: please can you check the colour for group 10, as it seems to be different between a and c?

*Thank you for highlighting this inconsistency. We have performed the change.*

(2) Figure 3: there seems to be one dot on the Canadian Shelf that has been assigned to group 4 when it looks as if it maybe more appropriate to be assigned to group 1?

*Thank you for your comment. It was an error in the file. It has been corrected.*

(3) Figure 4 caption: don't think need '(on a per one basis)'

*Deleted.*

(4) Figure 4 caption: 'Consult Table 1. . ..' rather than 'Confront Table 1. . ..'

*Done.*

(5) Figure 5 caption: 'Consult Table 1. . ..' rather than 'Confront Table 1. . ..'

*Done.*

(6) Figure 5 caption: Add sentence about what error bars are

*Thank you for your suggestion. We have added the following text to Figure 5 caption: "Error bars represent the error in the net water mass volume transport for 2014 and the standard deviation from the average net water mass volume transport for 2002–2010".*

(7) Figure 5: consider moving IrSPMW down, so have all water masses that contribute to upper limb of AMOC, and then all water masses that contribute to lower limb (?)

*Thank you for your suggestion. IrSPMW has been moved between $NEADW_L$ and PIW.*

(8) Figure 6 caption: don't think need '(on a per one basis)'

*Deleted.*

(9) Figure 6 caption: 'Consult Table 1. . ..' rather than 'Confront Table 1. . ..'

*Done.*

(10) Figure 6: consider using different colour-scale e.g. one that has white around 0, warm colours for positive anomalies and cool colours for negative anomalies.

*Thank you for your comment. We have changed the color scale following your suggestion.*

(11) Figure S1: please make axis lines thicker

*Thank you for your suggestion. We have increased the axis thickness.*

(12) Figure S1: what are the units for plot a?

*The total residual from the OMP is dimensionless. The total residual is the value of the squared 2-norm of the residual of the linear least square equations.*

(13) Figure S2: consider using different colour-scale e.g. one that has white around 0, warm colours for positive anomalies and cool colours for negative anomalies.

*Thank you for your comment. We have changed the color scale following your suggestion.*

***English / Typo suggestions:***

(1) P2, L7: insert word '...that enable us to trace back...'

*Done.*

(2) P2, L15: insert word '.... as in the subpolar...'

*Done.*

(3) P2, L22: replace 'in' with 'of' '... consisted of 78...'

*Done.*

(4) P2, L23-24: don't need to mention Short, Large, etc stations.

*The sentence has been deleted.*

(5) P2, L25: mention what SBE43 is

*The information has been added.*

(6) P2, L25: insert word '...was used as a reference...'

*Done.*

(7) P2, L25: insert word '...reference for the physical...'

*Done.*

(8) P2, L27: replace 'in' with 'at' '...performed at all...'

*Done.*

(9) P2, L28: don't need CFA abbreviation as not used again in manuscript

*Deleted.*

(10) P2, L29: 'dividing' not 'diving'

*Done.*

(11) P2, L30: please give the accuracy for nutrients in µmol kg-1 rather than µM

*Done.*

(12) P3, L21: insert 'that' '... NEADW that can be...'

*Done.*

(13) P3, L33: don't use 'IcSPMW' anywhere else in the paper. Do you mean SPMW7?

*Thank you for making us notice this inconsistency. We have replaced 'IcSPMW' with 'SPMW$_7$'.*

(14) P4, L14: 'constraint' not 'constrain'

*Done.*

(15) P4, L22-25: don't think you need these lines. Mention it separating out biological and mixing components will be valuable when interpreting TEI distributions but then don't show in manuscript!

*Thank you for your comment. We have deleted those lines.*

(16) P5, L6-8: re-order words '...Iceland Basin, with the $\theta$-S of SPMW8 being representative of that formed within the Iceland Basin...and the $\theta$-S of SPMW7 to that found over the eastern...'

*Done.*

(17) P5, L18: 'crossed' not 'crossing'

*Done.*

(18) P5, L26: remove 'set'

*Done.*

(19) P5, L29: 'was' not 'were' '...NEADWU was...'

*Done.*

(20) P6, L6: change to '...allowed an assessment of...'

*Done.*

(21) P6, L19: change word order '... (Fig 4a.b), with ENACW12 being...'

*Done.*

(22) P6, L28: insert 'to' '…NAC leads to the formation...'

*Done.*

(23) P7, L4: change word order '. . ., with SAIW6 being the...'

*Done.*

(24) P7, L10: insert 's' '...LSW concentrations reaching...'

*Done.*

(25) P7, L11: insert 'the' '...from the surface...'

*Done.*

(26) P7, L15: change word order '... with the first 1000 dbar of the Irminger Sea being dominated...'

*Done.*

(27) P7, L16: 'mixture' not 'mixing'

*Done.*

(28) P7, L18: insert 'water' '... LSW-like water...'

*Done.*

(29) P7, L23:24: change word order 'Some authors refer to the admixture... found over and around the Reykjanes Ridge as Icelandic Slope Water…'

*Done.*

(30) P8, L22: 'hydrographic' rather than 'hydrological' ?

*Yes. We have replaced 'hydrographical' with 'hydrologic'.*

(31) P9, L5: insert words '... intensity of the AMOC...'

*Done.*

(32) P9, L24: insert 'our' 'Since our OMP analysis...'

*Done.*

(33) P9, L24: remove 'the' '... time invariant properties...'

*Done.*

(34) P10, L6: add 's' and 'it' '...greater depths it is LSW...'

*Done.*

(35) P10, L17: add 'the' '.. at the expense...'

*Done.*

(36) P10, L29: add 'with the' '(Fig 6e), with the redistribution...'

*Done.*

(37) P11, L7: change 'con la'

*Thank you for making us notice this mistake. We have replaced 'con la' with 'and'.*

(38) P11, L16: do the authors mean Greenland Slope rather than Greenland Shelf?

*Yes. We have replaced 'shelf' with 'slope'.*

(39) P12, L14: insert 'an' '... to an average...'

*We have replaced 'to an' with 'with the'.*

(40) P13, L1: change to '... allows identification of the water...'

    *Done.*

(41) P13, L20: insert 'with the' '...2002-2010, with the increase related...'

    *Done.*

(42) Abstract, P1, L19: insert 's' '...colder end-members of the...'

    *Done.*

(43) Abstract, P1, L21: insert 'with the' '... 2002-2010, with the increase...'

    *Done.*

(44) Abstract, P1, L25: 'identification of' not 'identifying'

    *Done.*

*References:*

*Johns, W., Houk, A., Koman, G., Zou, S., and Lozier, S.: Transport of Iceland-Scotland Overflow waters in the Deep Western Boundary Current along the Reykjanes Ridge, Geophysical Research Abstracts, 19, EGU2017-9415, 2017.*

*Leffanue, H., and Tomczak, M: Using OMP analysis to observe temporal variability in water mass distribution, Journal of Marine Systems, 48 (1), 3–14, doi:10.1016/j.jmarsys.2003.07.004, 2004.*

*Olsen, A., Key, R. M., van Heuven, S., Lauvset, S. K., Velo, A., Lin, X., Schirnick, C., Kozyr, A., Tanhua, T., Hoppema, M., Jutterström, S., Steinfeldt, R., Jeansson, E., Ishii, M., Pérez, F. F., and Suzuki, T.: The Global Ocean Data Analysis Project version 2 (GLODAPv2) – an internally consistent data product for the world ocean, Earth Syst. Sci. Data, 8, 297-323, https://doi.org/10.5194/essd-8-297-2016, 2016.*

*Yashayaev, I., and Loder J. W.: Further intensification of deep convection in the Labrador Sea in 2016, Geophys. Res. Lett., 44, 1429–1438, doi:10.1002/2016GL071668, 2017.*

*Zou, S., Lozier, S., Zenk, W., Bower, A., and Johns, W.: Observed and modeled pathways of the Iceland Scotland Overflow Water in the eastern North Atlantic, Progress in Oceanography, 159, 211–222, doi:10.1016/j.pocean.2017.10.003, 2017.*

*Zunino, P., Lherminier, P., Mercier, H., Daniault, N., García-Ibáñez, M. I., and Pérez F. F.: The GEOVIDE cruise in May-June 2014 revealed an intense MOC over a cold and fresh subpolar North Atlantic, Biogeosciences, 14, 5323-5342, doi:10.5194/bg-14-5323-2017, 2017.*

---

## Author Comment (AC2) · 27 Feb 2018

*We thank referee #3 for the helpful comments. We have addressed the referee's concerns as explained below.*

Detailed comments

I am not familiar with the content in the much cited Zunino et al., which might limit my ability to interpret some of the findings in this manuscript.

*Thank you for making us notice that we refer too much to Zunino et al. (2017) and the reader could be confused. According to your comments below, we have rewritten section 4.1, which describes and discusses the changes in the water masses between 2002-2010 and 2014. We have reduced the redundant citations to Zunino et al. (2017) and we have added more information when needed to better guide the reader.*

The use of a subscript (e.g. SPMW7) for different types the otherwise well known water masses, is new to me. And this notation is not even introduced in this manuscript. I must admit that this detail hindered me in following this manuscript initially. This confusion might be caused by my lack of knowledge, but this will probably confuse other readers as well. Please give a better introduction to this, and improve the integration with the literature. E.g. how does the SPMW7 associated with the water mass descriptions given in other oceanographic papers?

*We used the subscripts to denote that it is the same water mass but with slightly different temperature and salinity. In fact, the subscript indicates the temperature of the SWT. Similar notation was used in other works (e.g., Álvarez et al., 2004; van Aken and de Jong, 2012). However, García-Ibáñez et al. (2015) was the first OMP-based work to use different SPMW end-members, according to our knowledge. Therefore, there are no other works using this notation. However, we believe that the first paragraph on page 5 (copied below) that introduces the notation will help the reader to understand the notation.*

*"The upper waters of the GEOTRACES-GA01 section were characterised by Central Waters and SPMW. The thermohaline range of the Central Waters was solved by defining two SWTs that coincide with extremes of the $\theta$-S line defining the East North Atlantic Central Waters (ENACW), the predominant variety of the North Atlantic Central Waters to the east of the Mid-Atlantic Ridge (Iselin, 1936): ENACW of 16ºC (ENACW$_{16}$), whose $\theta$-S characteristics match those from the warmer central waters of Pollard et al. (1996); and ENACW of 12ºC (ENACW$_{12}$), which represents the upper limit of ENACW defined by Harvey (1982). The change in temperature of SPMW along the NAC path cannot be accounted by the OMP analysis, since it is the result of air-sea interaction (e.g., McCartney and Talley, 1982; Brambilla and Talley, 2008). This problem was solved by defining three SWTs to characterize SPMW: SPMW of 8ºC (SPMW$_8$), SPMW of 7ºC (SPMW$_7$) and SPMW of the Irminger Basin (IrSPMW). SPMW$_7$ and SPMW$_8$ characterize the thermohaline range of SPMW in the Iceland Basin, with the $\theta$-S of SPMW$_8$ being representative of that formed within the Iceland Basin (Brambilla and Talley, 2008); and the $\theta$-S of SPMW$_7$ to that found over the eastern flank of the Reykjanes Ridge (Thierry et al., 2008). The $\theta$-S of IrSPMW characterize SPMW found in the Irminger Sea (Brambilla and Talley, 2008), and are close to those of the Irminger Sea Water (Krauss, 1995). The intermediate waters of the GEOTRACES-GA01 section were characterised by LSW, MW and SAIW. The thermohaline properties of LSW were chosen from the thermohaline properties of LSW formed in 2008*

*(LSW2008; Kieke and Yashayaev, 2015; Yashayaev and Loder, 2009, 2017), which, according to the transit times proposed by Yashayaev et al. (2007), would have reach the Irminger and Iceland basins by 2014. The properties of MW were taken from Wüst and Defant (1936) near Cape St. Vicente, where MW has its $\theta$-S characteristics established after overflowing the Strait of Gibraltar (Ambar and Howe, 1979; Baringer and Price, 1997). The thermohaline range of SAIW (4–7ºC and S < 34.9) was represented by two SWTs: SAIW of 6ºC ($SAIW_6$) and SAIW of 4ºC ($SAIW_4$), following the descriptions of Bubnov (1968) and Harvey and Arhan (1988). Finally, the deep waters of the GEOTRACES-GA01 section were characterised by DSOW, ISOW and NEADW. The thermohaline properties of ISOW were defined as the ISOW properties after crossed the Iceland-Scotland sills defined by van Aken and Becker (1996), and were readjusted by increasing its temperature and salinity by 0.1ºC and 0.01, respectively, according to the observed changes in the overflow properties since 2002 (Hansen et al., 2016). The thermohaline characteristics chosen for DSOW were selected from those found by Tanhua et al. (2005) downstream of the Greenland-Iceland sill. We also included PIW in the analysis to take into account the dense shelf water intrusions into DSOW. The thermohaline characteristics selected for PIW are in agreement with those proposed by Malmberg (1972) and Rudels et al. (2002). NEADW was modelled by the definition of two SWTs equal to the end-points of the line defining the thermohaline properties of NEADW in the West European Basin (Saunders, 1986; Mantyla, 1994; Castro et al., 1998): upper NEADW ($NEADW_U$) and lower NEADW ($NEADW_L$)".*

This work seems to use different – and maybe lower - values for the nutrient concentrations in the SWTs, compared to some other studies. The authors e.g. use a silicic acid concentration of 6.33 µM to represent the MW, while (McGrath et al., 2012) use a silicate concentration of 10-11 µM for the same water mass.

*Note that the values selected as nutrient concentrations to characterize the SWTs are preformed values, that is, the values the water mass acquired when it was formed. That is the reason why the concentration of silicic acid for MW in our work differs from that reported by McGrath et al. (2012) for measurements further north from the formation area of MW.*

a) Is silicic acid, Si(OH4)0, not the same as "silicate"? b) Why use both one and two decimals in Table 1? c) How sensitive is eOMP method to such different choices of the source water silicic acid concentration?

*a) The notation "silicate" is commonly used instead of "silicic acid" for simplicity, but both notations denote the same. b) The number of decimals was set to show the accuracy, giving two decimals when the standard deviation was lower than 0.2. c) The importance of the silicic acid concentrations when solving the eOMP analysis is that it tracks NEADW, that is, the water masses with high silicic acid concentration. In the Irminger Basin and in the main thermocline, nitrate and oxygen are better tracers to solve the water mass distribution. We did not perform a Monte-Carlo simulation only perturbing the silicic acid values describing the SWTs to evaluate the sensitivity of the eOMP of the choice of silicic acid values for the SWTs. However, the Monte-Carlo simulation performed by perturbing all the physical and chemical properties defining the SWTs leads to an average standard deviation of distribution of SWTs lower than 12%, which indicates that the methodology is robust.*

I am a bit confused by the definition and discussion of the 'Central Water'. On page 5, line 26 (p5,l26) this water mass is defined as ENACW16+ ENACW12, on p5,l29 is stated that "The distribution of the Central Waters is associated with the NAC" and on (p3,l6) is stated that the Central Waters is transported with the NAC. How can it be defined by the eastern waters, and be transported with the NAC? Please clarify.

*Sorry for the confusion. To the east of the Mid-Atlantic Ridge in the North Atlantic, the predominant variety of the North Atlantic Central Waters (Iselin, 1936) is the East North Atlantic Central Water (ENACW) (Harvey, 1982; Pollard et al., 1996; Read, 2000), which is formed by winter convection in the intergyre region (Pollard et al., 1996). This is the reason why we chose the ENACW nomenclature to refer to the Central Waters. We added the following information when defining the water masses we used: "The thermohaline range of the Central Waters was solved by defining two SWTs that coincide with extremes of the $\theta$-S line defining the East North Atlantic Central Waters (ENACW), **the predominant variety of the North Atlantic Central Waters to the east of the Mid-Atlantic Ridge (Iselin, 1936)**: (…)".*

(p5,l6): "An important assumption of the methodology is that the physical and chemical characteristics of the SWTs are considered time invariant and..." (p9,l31): "...the progressive salinization that classical LSW (our SWT) has been experiencing since its last formation event in the late 1990s..." So the eOMP method seems to be importantly dependent on the assumption of time invariability of the SWTs, and it is clear that the SWT are not time invariable. At first glance, this appears as a contradiction. Please explain.

*We are aware that the properties of LSW and ISOW have been changing over time. To take this fact into account and following the comments of referee #1, the temperature and salinity (TS) for LSW and ISOW have been slightly modified compared to García-Ibáñez et al. (2015) to match those found in the most recent period. We have also revised the standard deviations of the properties that define the SWTs taking into account the temporal variability. The TS properties for LSW in this new run are 3.4ºC and 34.855, thermohaline properties chosen from LSW formed in 2008 (LSW$_{2008}$, Kieke and Yashayaev, 2015, Yashayaev and Loder, 2009, 2017), which, according to the transit times proposed by Yashayaev et al. (2007), would have reached the Irminger and Iceland basins by 2014. The TS properties for ISOW in this new run are 2.7ºC and 35, that is, an increase in temperature of 0.1ºC and an increase in salinity of 0.01, according to the changes observed in the overflow properties since 2002 (Hansen et al., 2016). We have used the results of the new OMP run as the final results of the manuscript. By making this change, we believe that the contradiction is resolved. Besides, the salinization of LSW to which we refer is due to lateral mixing of LSW with surrounding waters once formed, and not to the salinization of its source area.*

(p7,l32): "...measurements and by an overall mass balance of 1 ± 3 Sv northwards..." It is not clear what this means.

*We have changed the text: "… and by  **a net volume transport** of 1 ± 3 Sv northwards **to ensure mass conservation**".*

(p9,l13) and below: The abbreviation SMPW is often used. I assume this should be SPMW. Section 4.1. The discussion on the water mass changes between the average 2002-2012 state, and 2014 is difficult to follow. This is partly because Fig. 6 could be improved (see comments below), and partly because the patterns are not always clear. Maybe guide the reader better to the mentioned changes (e.g. specify depths levels).

*Thank you for highlighting the mistake in the acronym SPMW. We have made the replacement. Regarding the discussion about the water mass changes, we have improved Fig. 6, following your suggestions and the comments from referee #1. We have also rewritten section 4.1 to improve the message.*

(p9,l27): "The negative anomalies of LSW between 1000 and 2000 dbar coincide with positive anomalies of SPMW7..." It makes sense that the cooling after 2014 was associated with a replacement of the relatively warm SPMW7 with the colder LSW. But it seems counterintuitive that the opposite occurred below 1000 dbar. Was LSW really replaced by the warmer SPMW at these deeper levels? Please explain. One result of this paper is an unexpectedly high presence of ISOW. It is known that the eOMP is sensitive to the assumption of time invariability of the SWTs, and it is clear that the ISOW SWT became more saline after 2002. Could the unexpectedly high presence of ISOW partly be a result of this uncertainty?

*We are aware that the temperature and salinity (TS) of some water masses have changed over time, e.g. LSW and ISOW, and, therefore, we have performed a new OMP run with the TS properties defining LSW and ISOW slightly modified in relation to those used in García-Ibáñez et al. (2015) to match those found in the most recent period (see answer to comment on p5, L6). Even the results of the new OMP run show proportions of ISOW higher than the mean values reported in the literature. Therefore, we are confident that the higher than expected concentration of ISOW is a real feature, which is in agreement with the volume transport of ISOW observed in the OSNAP array (Johns et al., 2017; Zou et al., 2017). We have added this fact in the manuscript: "The uniform increase in ISOW is consistent with the increase in volume transport of ISOW observed in the OSNAP array (Johns et al., 2017; Zou et al., 2017)". Besides, when using the results of the new OMP run, the replacement of LSW by SPMW$_7$ below 1000 dbar disappears, being LSW replaced by ISOW, which is more consistent.*

Figures

Figures 1-5 are all ok.

The NAC in Fig. 1 is located farther south than where we usually see it (in the literature). Is this because the authors suggest that the NAC is actually located this far south?

*The figure represents average location of the NAC during 2002-2012 based on the OVIDE and 60°N sections (Daniault et al., 2016). The location of the NAC branches at the Mid Atlantic Ridge is from Bower and von Appen (2008) and they are locked to the Charlie–Gibbs Fracture Zone, Faraday Fracture Zone and Maxwell Fracture Zone. The location of the last two fracture zones has been added to Fig. 1.*

Fig. 2d. The silicic acid values span 0-40 µM, probably in order to get the highest values near the seafloor in the eastern part represented as well. But most of the observed silicate variability is seen in the range 2-12 µM, and the figure has a low resolution in this range. Maybe consider using a non-linear color code?

*Thank you for the suggestion. We have changed the color code to be non-linear.*

Figure 6

The message in this figure is not clear.

a) Maybe use different software. Although ODV is well suited to scan oceanographic data, it might not be the right choice for making publishable figures. If you still want to use ODV, remove the redundant references to this software.

*Thank you for the suggestion. We prefer to continue using ODV. We cannot do anything about the redundant references to the software because ODV does not allow removing the references for each plot. We have improved the figure by changing the color scale to one that has white around zero, warm colors for positive values and cool colors for negative values, as suggested by referee #1.*

b) Use the y-axis label, "Pressure (dbar)" only once. The same goes for the other ODV-based figures (Figs. 2 and 4).

*Thank you for the suggestion. We have performed the suggested change in Figures 2, 4 and 6.*

c) What does "(on a per one basis)" really mean?

*It means that the proportion of each SWT is represented ranging from 0 to 1. Following the suggestion of referee #1, we have deleted 'on a per one basis' from the figure captions.*

d) Maybe add something to the text in (p9,l4) "Positive (negative) anomalies in the proportion of a water mass imply a gain (loss) in 2014 compared to 2002–2010." to the caption for Fig. 6.

*Thank you for your suggestion. We have added that text to Figure 6's caption.*

e) Since the patterns in this figure are quite noisy, one can doubt the usefulness of this figure. The uncertainty about the parameter shown, and the definition of the water types with that subscript (e.g. ENACW16, see comment above), it becomes difficult to follow the discussion related to this figure. Suggestions: a) Show and discuss only the clearest signals (fewer panels in the figure). Patterns with blue and red blobs might too much associated with the inherent variability, which could strongly impact a single transect along the OVIDE line. Or b), improve the figure and the explanation of its content, and integrate the discussion with this figure in a clearer way.

*Thank you for your suggestions. Following your suggestion, we have reduced the number of subplots, showing only those water masses with clear patterns. We have also rewritten the section explaining this figure to improve the message.*

References

McGrath, T., Nolan, G., McGovern, E., 2012. Chemical characteristics of water masses in the Rockall Trough. Deep-Sea Research Part I-Oceanographic Research Papers 61, 57-73.

*References:*

*Álvarez, M., Pérez, F. F., Bryden, H. and Ríos, A. F.: Physical and biogeochemical transports structure in the North Atlantic subpolar gyre, J. Geophys. Res., 109, C03027, doi:10.1029/2003JC002015, 2004.*

*Bower, A. S., and von Appen W. J.: Interannual variability in the pathways of the North Atlantic current over the Mid-Atlantic Ridge and the impact of topography, Journal of Physical Oceanography, 38, 104-120, doi:10.1175/2007JPO3686.1, 2008.*

*Daniault, N., Mercier, H., Lherminier, P., Sarafanov, A., Falina, A., Zunino, P., Pérez, F.F., Ríos, A.F., Ferron, B., Huck, T., Thierry, V., and Gladyshev, S.: The northern North Atlantic Ocean mean circulation in the early 21st century, Progress in Oceanography, 146, 142-158, doi:10.1016/j.pocean.2016.06.007, 2016.*

García-Ibáñez, M. I., Pardo, P. C., Carracedo, L. I., Mercier, H., Lherminier, P., Ríos, A. F., and Pérez F. F.: Structure, transports and transformations of the water masses in the Atlantic Subpolar Gyre, Progress in Oceanography, 135, 18–36, doi:10.1016/j.pocean.2015.03.009, 2015.

Hansen, B., Húsgarð Larsen, K. M., Hátún, H., and Østerhus, S.: A stable Faroe Bank Channel overflow 1995–2015, Ocean Sci., 12, 1205-1220, doi:10.5194/os-12-1205-2016, 2016.

Harvey, J.: Theta-S relationships and water masses in the eastern North Atlantic, Deep Sea Research Part A: Oceanographic Research Papers, 29 (8), 1021–1033, doi:10.1016/0198-0149(82)90025-5, 1982.

Iselin, C.O.: A Study of the Circulation of the Western North Atlantic, Pap. Phys. Oceanogr. Meteorol. Massachusetts Inst. Tech. and Woods Hole Oceanographic Inst, 101p, 1936.

Johns, W., Houk, A., Koman, G., Zou, S., and Lozier, S.: Transport of Iceland-Scotland Overflow waters in the Deep Western Boundary Current along the Reykjanes Ridge, Geophysical Research Abstracts, 19, EGU2017-9415, 2017.

Kieke, D., and Yashayaev, I.: Studies of Labrador Sea Water formation and variability in the subpolar North Atlantic in the light of international partnership and collaboration, Progress in Oceanography, 132, 220–232, doi:10.1016/j.pocean.2014.12.010, 2015.

Pollard, R. T., Grifftths, M. J., Cunningham, S. A., Read, J. F., Pérez, F. F., and Ríos, A. F.: Vivaldi 1991 – a study of the formation, circulation and ventilation of Eastern North Atlantic Central Water, Progress in Oceanography 37, 167–192, doi:10.1016/S0079-6611(96)00008-0, 1996.

Read, J. F.: CONVEX-91: water masses and circulation of the Northeast Atlantic subpolar gyre, Progress in Oceanography, 48 (4), 461–510, doi:10.1016/S0079-6611(01)00011-8, 2000.

van Aken, H.M., and de Jong, M.F.: Hydrographic variability of Denmark Strait Overflow Water near Cape Farewell with multi-decadal to weekly time scales, Deep Sea Research Part I: Oceanographic Research Papers, 66, 41–50, doi:10.1016/j.dsr.2012.04.004, 2012.

Yashayaev, I., and Loder J. W.: Enhanced production of Labrador Sea Water in 2008, Geophysical Research Letters, 36, L01606, doi:10.1029/2008GL036162, 2009.

Yashayaev, I., and Loder J. W.: Further intensification of deep convection in the Labrador Sea in 2016, Geophys. Res. Lett., 44, 1429–1438, doi:10.1002/2016GL071668, 2017.

Yashayaev, I., Bersch, M., and van Aken, H. M.: Spreading of the Labrador Sea Water to the Irminger and Iceland basins, Geophysical Research Letters, 34 (10), L10602, doi:10.1029/2006GL028999, 2007.

Zou, S., Lozier, S., Zenk, W., Bower, A., and Johns, W.: Observed and modeled pathways of the Iceland Scotland Overflow Water in the eastern North Atlantic, Progress in Oceanography, 159, 211–222, doi:10.1016/j.pocean.2017.10.003, 2017.

---

## Author Comment (AC3) · 27 Feb 2018

*We appreciate the stimulating comments from referee #2. We have replied to his/her comments below.*

The first reviewer has already provided a summary of the paper, so I will just go to straight to my points. However, everything else I say below solely reflects my opinion and view on the complex process of water mass formation and variability in the North Atlantic.

The issues the authors address in the paper are highly relevant and important for water mass analysis and prediction of their changes over time, and dissecting transformation and mixing of water mass is a big and nontrivial problem overall, so any novel solid approach and a study based on it would be much awaited here.

However, I cannot understand how a trans-Atlantic snapshot (not to mention that the section does not end in St. John's, Newfoundland) and a simple model operating with only four members at once can be used to depict complex interaction and mixing of 14 water masses. I am not in position to judge the previously published paper of the same authors that is used as a basis for the current one, but if I had to review it, I would come with critical suggestions pretty much similar to those presented below.

Let me explain why I believe that a four-member approach does not work for this specific task:

(1) First of all - the case is not two-dimensional (2D distance along section vs depth). The water masses interact in over the entire subpolar North Atlantic. So, for example, any two waters appearing as neighbors on the OVIDE line may be separated by other waters elsewhere in the region. Therefore, the only way to solve this problem for the subpolar North Atlantic and its water masses is through solving a full system of equations where each end-member is careflly defined, and this creates another challenge.

*We agree that the water mass circulation, formation and transformation in the subpolar North Atlantic is a complex problem to solve. That is the reason why we carefully defined the SWT properties in their formation area based on values available in the literature. We would also like to emphasize that OMP offers tools to verify the assumptions made and the consistency of the analysis. Most importantly, we verified that the residuals of the OMP equations were small enough to conclude that all samples could be described by the mixing of selected SWT (p6 L4-13): "We tested the robustness of the methodology through a Monte-Carlo simulation (Tanhua et al., 2005), where the physical and chemical properties of both each SWT and each water sample were randomly perturbed within the standard deviation of each parameter (see Text S1 and Table S1). This allowed an assessment of the sensitivity of the eOMP analysis to potential measurement errors and temporal variations in the physical and chemical properties that define the SWTs (Leffanue and Tomczak, 2004). A hundred Monte-Carlo simulations were performed and the eOMP equation system was solved for each of them. The average standard deviation of the $X_i$s (last column in Table 1) is lower than 12%, which indicates that the methodology is robust. Additionally, our eOMP analysis is consistent since its residuals (r in Eq. 3) lack a tendency with depth (Fig. S1), with the standard deviations of the residuals being slightly higher than the measurement errors (Table 1). Besides, the ability of out eOMP analysis to reproduce the measured values is given as the correlation coefficient ($R^2$, Table 1) between the measured values (water samples) and the expected values for the SWT mixing (values of the properties of each water sample obtained by when substituting $X_i$s in*

*Eq. (3)). The R² values are higher than 0.993, which again indicates the reliability of our eOMP analysis".*

(2) Now, a whole list of problems concerning the end members: a) The authors use end-member properties as they appear on a snapshot of an arbitrary section line (OVIDE or any) and not the properties of the studied water masses that these waters acquired at the times of their formation. Most critically here, both DSOW and ISOW should start from sub-zero temperatures. Both ISOW and DSOW are equally fresh the sills. However, ISOW gains its salt through mixing as it spread through the Iceland Basin. So taking the water that is already salty is not good for telling how it was formed from start – note that it has already been mixed with SPMW. Same is true about the other waters.

*We understand your concern. As indicated in page 5, lines 1-23 from the submitted manuscript, the properties selected to define the SWTs (end-members) were taken from the literature, from the regions where the water masses are formed and were not an arbitrary selection to fit the properties of the GEOVIDE cruise. In the case of the overflow waters, we considered that these water mases are formed once they had crossed the sills, that is, once they overflowed and entrained ambient waters. Solving the composition of ISOW and DSOW themselves is not the aim of the present manuscript, and could not be done with the data from the GEOVIDE cruise. We believe that this approach is legitimate because those characteristics of ISOW and DSOW (i.e., after the overflow process) are the most commonly used to track the overflows in the Atlantic Ocean (e.g., Dickson et al., 2002; Fogelqvist et al., 2003; Tanhua et al., 2005;2008; Yashayaev and Dickson, 2008).*

(b) By no means, LSW remains undiluted between Labrador Sea and Iceland Basin. However, Figure 4 suggests 90% of original LSW in any other LSW all the way through the region. Well, the Labrador Sea is a very powerful engine, but can it pump so much water that stays unmixed for so far and so long?

*In fact we observed LSW proportions up to 90%, but the bulk of it is found in the Irminger Sea, which is another proposed area of LSW-like formation (e.g., de Jong and de Steur, 2016; Piron et al., 2017). Besides, LSW is up to 2000 m thick in the Labrador Sea and, therefore, it is not that surprising that its core characteristics experience little change (about 0.1°C in temperature and 0.01 in salinity as shown in, for example, Yashayaev et al., (2007)) while being advected towards the Iceland Basin. This explains the high proportion of LSW found in the Iceland Basin at the core of the water mass.*

(c) The depth of LSW was not 2000 m in 2014, and there cannot 50% of LSW at 3000 m – at the depths where water is already as saline as ISOW modified through entrainment.

*We agree that the deep convection in the Labrador Sea was not as deep as in the late 1990s. However, our distribution of LSW in the Labrador Sea could reflect the diapycnal mixing of LSW with ISOW or the entrainment of LSW into the ISOW layer all around the subpolar gyre. In his talk at Ocean Sciences 2018, Bill Johns showed that ISOW is modified through entrainment of LSW on its way along the eastern flank of the Reykjanes Ridge downstream of the Iceland-Scotland sill. High diapycnal mixing has been observed in the Deep Western Boundary Current in the Irminger Sea by Lauderdale et al. (2008). Entrainment and mixing would explain finding high percentages of LSW at depth, associated with the circulation of ISOW. We have added the following information in the text to explain the distribution of LSW in deep layers of the Labrador Sea:* "The distribution of LSW in the Labrador Sea that extends deeper than 2000 dbar reflects the diapycnal mixing with ISOW

*(Lauderdale et al., 2008) and/or the entrainment of LSW in the ISOW layer all along the subpolar Gyre". However, we would be very interested in evidences that show that there cannot be 50% of LSW at 3000 m.*

(d) Then, ISOW is fresher in Labrador Sea than in the Irminger Sea, because it is more diluted, but the corresponding fractions seem very much comparable in Figure 4. Does ISOW really reach 2000 m in the Labrador Sea adding about 50%? Or is it something else? How can we be so sure that another water mater contributing to the mid-depth exchanges and arriving from outside the Labrador Sea is not missed in this formulation? It must be something else rather than 50% of ISOW...

*According to the general knowledge, the Deep Western Boundary Current transports ISOW to the Labrador Sea (e.g., Schmitz and McCartney, 1993; Rudels et al., 2002; Tanhua et al., 2008), where it circulates cyclonically (Xu et al., 2010). We are not aware that this general view has been recently questioned. Our water mass distributions are in agreement with that circulation. In addition, the residuals of the OMP equations are small enough to conclude that all samples can be described by the mixing of the selected SWTs. On the other hand, we have not found in the literature more water masses described in the Labrador Sea. However, we would be very interested in the knowing the existence of another possible water mass contributing to the mixing in the Labrador Sea.*

(e) I totally agree that a more careful approach is needed for the two chemical variables used in the work. However, using a certain universal model for utilization may lead to overconsumption of oxygen at greater depth. I say this, because the oxygen section suggests weak biological utilization, whereas applying parameterizations used in biochemistry (I cannot expand further here, but any quick assessment would show a comparable result) would reduce dissolved oxygen more than what we see in the section. If we assume a strong bio-consumption, then how would we explain that dissolved oxygen closely follows salinity which in turn is not altered by living organisms?

*We do not see how is it possible to determine oxygen consumption rates from an oxygen section without explicitly calculating it. AOU is not negligible along our section. Our OMP setting is adequate to explain the complex mixing of water masses and ventilation/respiration processes that occur in the section, as evidenced by the low residuals and the fact that the selected SWTs and their mixing explain 99.99% of the temperature of the section, 99.9% for salinity, 99.4% for oxygen, 99.9% for silicic acid and 99.4% for nitrate. Regarding the similarity between the oxygen and salinity sections, this is mainly due to the circulation of MW. MW is characterized by high salinity and low oxygen concentration, the latter one related to the high temperature of this water mass.*

So far I was talking about using static end members assuming the picture does not change with time. But there is another set of complications coming into play if we introduce temporal variability of water properties. Yes, the source waters change in time, but any static model assumes invariance of the source waters. How long does it take for LSW to cross the basin? Let's say N years? How would the authors introduce the temporal changes previously observed in the source or sources of LSW? Note that convection was not strong in 2010 and 2011, and that it was that water that had probably been seen in the Iceland Basin in 2014! LSW does change a lot in its source in 3-4 years. How would this knowledge be transpired into 3.00 and 34.87 with such narrow error bars? At the season of formation the waters are even more different. Oxygen saturation is probably >95%. Taking the transit time into consideration, the version of LSW seen on the OVIDE line in the southern Labrador Sea may not be directly

related to that transferred to Iceland basin first through DWBC and then under NAC... The properties of the original waters can be much greater than the error bars used through the work. I bring LSW only as an example but the same may true about other waters brought into the equations.

*LSW represents a continuum of vintages whose properties do not change so dramatically from one year to the next (Yashayaev and Loder, 2016). In addition, the LSW vintages mix with the vintages from previous years both in the Labrador Sea and along its way to the Iceland Basin. Therefore, the signal of a concrete vintage is diluted, but the LSW entity itself is conserved due to the large volume that this water mass represents in the Subpolar Gyre. Therefore, only consecutive and persistence changes are observed far from its formation area (Sy et al., 1997). On the other hand, following the comments of referees #1 and #3, we have performed a new OMP run where we slightly modified the temperature and salinity (TS) properties for LSW and ISOW to match those found in the most recent period. The TS properties for LSW in this new run are 3.4ºC and 34.855, thermohaline properties chosen from LSW formed in 2008 (LSW$_{2008}$, Kieke and Yashayaev, 2015, Yashayaev and Loder, 2009, 2017), which, according to the transit times proposed by Yashayaev et al. (2007), would have reached the Irminger and Iceland basins by 2014. The TS properties for ISOW in this new run are 2.7ºC and 35, that is, an increase in temperature of 0.1ºC and an increase in salinity of 0.01, according to the changes observed in the overflow properties since 2002 (Hansen et al., 2016). We have also revised the standard deviations of the properties that define the SWTs taking into account the temporal variability (you can see the new STDs in Table 1, copied below):* "**Text S1**

*The standard deviations (STD) of the potential temperature and salinity that define the source water types (SWTs) were taken from the literature. For Central Waters and SPMWs, the STDs were set as ± 0.6ºC for temperature and ± 0.06 for salinity, according to the thermohaline variability reported by Robson et al. (2016) for the first 700 m of the water column of the subpolar gyre. For LSW, the STDs were set as ± 0.4ºC for temperature and ± 0.01 for salinity, to include both the thermohaline properties used in García-Ibáñez et al. (2015) and those used in this work. For SAIW, the STDs were set as ± 0.5ºC for temperature and ± 0.03 for salinity, based on the variability of the thermohaline of its source waters, i.e., Central Waters and LSW (Iselin, 1936; Arhan, 1990; Read, 2000). For MW, the STDs were set as ± 0.2ºC for temperature and ± 0.07 for salinity, according to the work of Carracedo et al. (2016). For ISOW, the STDs were set as ± 0.1ºC for temperature and ± 0.02 for salinity, to include both the thermohaline properties used in García-Ibáñez et al. (2015) and those used in this work. For DSOW, the STDs were set as ± 0.16ºC for temperature and ± 0.008 for salinity, according to the work of Jocchumsen et al. (2012). For PIW, the STDs were set as ± 0.2ºC for temperature and ± 0.03 for salinity, according to the work of Falina et al. (2012). For NEADW$_L$, the STDs were set as ± 0.03ºC for temperature and ± 0.003 for salinity, according to the work of García-Ibáñez et al. (2015). For NEADW$_U$, the STDs for potential temperature and salinity were calculated using the STDs of its components: MW, LSW, ISOW and NEADW$_L$ (Sect. 2.3 of the main text).*

*For oxygen, the STDs were set equal to 3% of the saturation value (Najjar and Keeling, 2000; Ito et al., 2004), whereas for nutrients they were obtained by one of the following methods:*

*a) For to LSW, ISOW and NEADW$_L$, the STDs for the nutrients was calculated using the STDs in the water samples with more than 95% of those SWTs, following Karstensen and Tomczak (1998). This method was used when the number of water samples for a SWT was greater than 50.*

*b) For the Central Waters, DSOW and SPMW, which are defined by more than one SWT (multi-SWTs), the multi-SWT contributions were obtained by adding the contributions of their respective components. Then, water samples with proportions of the multi-SWT greater than 95% were selected. The property values of each component of the multi-SWT were subtracted from the values of the water samples and linear regressions were performed between potential temperature and nutrients. The STDs of the multi-SWT nutrients were taken equal to the error*

*of the intercept. We used the STDs of the properties of the multi-SWTs to each of their components.*

c) *A modification of the methodology (b) was applied to MW, where samples with proportions greater than 75% were selected to perform the linear regressions.*

*The STDs of the nutrients of SAIW were assigned equal to those of the Central Waters, because not enough water samples presented proportions greater than 95%. The STDs of the nutrients of $NEADW_U$ were calculated using the STDs of its components: MW, LSW, ISOW and $NEADW_L$ (Sect. 2.3 of the main text).*

*References:*

*Carracedo, L. I., Pardo, P. C., Flecha, S., and Pérez, F. F.: On the Mediterranean Water Composition, J. Phys. Oceanogr., 46, 1339–1358, https://doi.org/10.1175/JPO-D-15-0095.1, 2016.*

*Ito, T., Follows, M. J., and Boyle, E. A.: Is AOU a good measure of respiration in the oceans?, Geophysical Research Letters, 31, L17305, doi:10.1029/2004GL020900, 2004.*

*Jochumsen, K., Quadfasel, D., Valdimarsson, H., and Jónsson, S.: Variability of the Denmark Strait overflow: moored time series from 1996–2011, Journal of Geophysical Research, 117, C12003, doi:10.1029/2012JC008244, 2012.*

*Najjar, R.G., and Keeling, R.F.: Mean annual cycle of the air-sea oxygen flux: a global view, Global Biogeochemical Cycles, 14 (2), 573–584, doi:10.1029/1999GB900086, 2000".*

*The perturbation around the new STDs generates uncertainties in the proportions of the different SWTs lower than 12% (Table 1), which indicates that the methodology is robust against the temporal variability in the properties that define the SWTs.*

*Table 1: Properties characterising the Source Water Types (SWTs, see footnote a) considered in this study with their corresponding standard deviations[b]. The square of correlation coefficients ($R^2$) between the observed and estimated properties are also given, together with the Standard Deviation of the Residuals (SDR) and the SDR/ε ratios from the data below 400 dbar. The ε (standard deviation of the water sample properties) used to compute the SDR/ε ratios are listed in Table S1. The last column accounts for the uncertainties in the SWTs contributions.*

| | $\Theta$ (ºC) | S | $O_2{}^0$ ($\mu mol\ kg^{-1}$) | $Si(OH_4)^0$ ($\mu mol\ kg^{-1}$) | $NO_3{}^0$ ($\mu mol\ kg^{-1}$) | Uncertainty |
|---|---|---|---|---|---|---|
| $ENACW_{16}$ | 16.0 ± 0.6 | 36.20 ± 0.06 | 246 ± 7 | 1.87 ± 0.12 | 0.00 ± 0.15 | 9% |
| $ENACW_{12}$ | 12.3 ± 0.6 | 35.66 ± 0.06 | 251 ± 8 | 1.3 ± 0.9 | 8.0 ± 1.1 | 10% |
| $SPMW_8$ | 8.0 ± 0.6 | 35.23 ± 0.06 | 289 ± 9 | 2.7 ± 1.9 | 11.4 ± 1.3 | 11% |
| $SPMW_7$ | 7.1 ± 0.6 | 35.16 ± 0.06 | 280 ± 8 | 5.20 ± 0.15 | 12.83 ± 0.15 | 6% |
| $IrSPMW$ | 5.0 ± 0.6 | 35.01 ± 0.06 | 310 ± 9 | 5.9 ± 0.4 | 14.1 ± 0.4 | 12% |
| $LSW$ | 3.40 ± 0.4 | 34.86 ± 0.01 | 307 ± 9 | 6.9 ± 0.7 | 14.8 ± 0.7 | 10% |
| $SAIW_6$ | 6.0 ± 0.5 | 34.70 ± 0.03 | 297 ± 9 | 6.0 ± 2.4 | 13.3 ± 1.2 | 9% |
| $SAIW_4$ | 4.5 ± 0.5 | 34.80 ± 0.03 | 290 ± 9 | 0.0 ± 2.4 | 0.0 ± 1.2 | 3% |
| $MW$ | 11.7 ± 0.2 | 36.50 ± 0.07 | 190 ± 6 | 6.30 ± 0.15 | 13.2 ± 0.2 | 2% |
| $ISOW$ | 2.7 ± 0.1 | 35.00 ± 0.02 | 294 ± 9 | 11.8 ± 0.9 | 14.0 ± 0.6 | 9% |
| $DSOW$ | 1.30 ± 0.2 | 34.905 ± 0.01 | 314 ± 9 | 7.0 ± 0.5 | 12.9 ± 0.8 | 7% |
| $PIW$ | 0.0 ± 0.2 | 34.65 ± 0.03 | 320 ± 10 | 8.4 ± 2.5 | 13.4 ± 1.2 | 9% |
| $NEADW_U$ | 2.5 ± 0.5 | 34.940 ± 0.07 | 274 ± 8 | 29.4 ± 0.6 | 18.1 ± 0.5 | [c] |
| $NEADW_L$ | 1.98 ± 0.03 | 34.895 ± 0.003 | 252 ± 8 | 48.0 ± 0.3 | 22.0 ± 0.5 | 3% |
| $R^2$ | 0.9999 | 0.9984 | 0.9939 | 0.9978 | 0.9941 | |
| SDR | 0.009 | 0.005 | 2 | 0.4 | 0.2 | |
| SDR/ε | 2 | 2 | 2 | 1 | 1 | |

[a]*$ENACW_{16}$ and $ENACW_{12}$ = Eastern North Atlantic Central Water of 16ºC and 12ºC, respectively; $SPMW_8$, $SPMW_7$ and IrSPMW = Subpolar Mode Water of 8ºC, 7ºC and of the Irminger Sea, respectively; LSW = Labrador Sea Water; $SAIW_6$ and $SAIW_4$ = Subarctic Intermediate Water of 6ºC and 4ºC, respectively; MW = Mediterranean Water; ISOW = Iceland–Scotland Overflow Water; DSOW = Denmark Strait Overflow Water; PIW = Polar Intermediate Water; and $NEADW_U$ and $NEADW_L$ = North East Atlantic Deep Water upper and lower, respectively.*
[b]*The standard deviation of the properties of the SWTs were obtained following the method described in the Supplementary Information (Text S1).*
[c]*No uncertainty is given for $NEADW_U$ since it is was decomposed between MW, LSW, ISOW and $NEADW_L$ (see Sect. 2.3).*

Is it really true that DSOW has no LSW mixed into it? I find it strange because in the northern Imringer Sea DSOW is cascading down the slope entraining both NEADW (ex-ISOW) and LSW and warmer waters.

> *It is true that DSOW mixes with LSW, NEADW and Atlantic waters when cascading the Greenland-Iceland sill. This mixing was taken into account by defining the properties of DSOW after the overflow process, since we assume that DSOW is formed when the water has crossed the sill, like other authors do (e.g., Fogelqvist et al., 2003; Tanhua et al., 2005; Yashayaev and Dickson, 2008). This point is included in the text by the following statements: "DSOW forms after the deep waters of the Nordic Seas overflow the Greenland–Iceland sill and entrain Atlantic waters (SPMW and LSW) (Read, 2000; Yashayaev and Dickson, 2008). […] The thermohaline characteristics chosen for DSOW were selected from those found by Tanhua et al. (2005) downstream of the Greenland-Iceland sill". In addition, one of our mixing groups (mixing group 3) allows the mixing of DSOW, PIW, LSW and ISOW to account for any additional mixture of LSW and DSOW downstream of the sill.*

The Monte Carlo technique would only help if the errors were random respecting central tendency. I have no doubt that each of the linear 4-member solutions would converge even with larger seeded errors. However, the present case is subject to more systematic rather than random biases, raising a question like "How would each solution change if LSW was 0.3 warmer at time of formation?"

> *The Monte Carlo technique has been commonly used to test the robustness of the OMP analysis for temporal variations in the properties of the end-members (e.g., Tanhua et al., 2005; Jeansson et al., 2008; Pardo et al., 2012). Besides, the residuals of the OMP analysis in terms of error in salinity, temperature, oxygen and nutrients do not generate any bias in relation to the SWT proportions. Therefore, the Monte Carlo technique is suitable to test the robustness of the selected SWT. However, it is true that the variability introduced in the properties that define the SWTs is less than the long-term variability of LSW properties, for example. That is why, as stated before, we have revised the standard deviations of the properties that define the SWTs taking into account the temporal variability, resulting in uncertainties of the water mass proportions lower than 12%, thus confirming the robustness of the method.*

Saying that the task of unscrambling water mass composition in this highly dynamic and variable area is well worth pursuing, I, unfortunately, cannot agree that the presented method, data and results help much solving this task. There must be a solution, but based on a more extensive synthesis of three-dimensional (3D) data, on a proper definition of source waters and their changing properties, on accounting for multiple pathways, etc.

> *We agree that the North Atlantic circulation, and water mass formation and transformation is a complex problem. One of the main criticisms to the OMP analysis is that its results are sensitive to the number and definition of the SWTs, and that the analysis is limited to distinguishing only as many water masses as there are distinct water properties. However, in our study we performed an OMP analysis using 14 SWTs defined by five properties. In order to solve an over-determined system, the SWTs were organized into 11 subsets of maximum four SWTs each, which were set according to the characteristics and/or dynamics of the water masses in the Subpolar North Atlantic. These 11 subsets were vertically and*

*horizontally sequenced, and they share at least one SWT with the adjacent subsets to ensure water mass continuity. This methodology allowed us to solve the complex water mass circulation in the section, the OMP results being consistent with the water mass circulation in the Subpolar North Atlantic, and explaining more than 99% of the properties observed in the section. Therefore, we argue that the OMP method is suitable for the study water mass distributions, and their formation and transformation (e.g. Álvarez et al., 2004; Tanhua et al., 2005;2008; Jeansson et al., 2008; Pardo et al., 2012; Carracedo et al., 2016), and it allows distinguishing the relative importance of conservative mixing from non-conservative processes on tracer distributions (e.g. Llanillo et al., 2013, de la Paz et al., 2017).*

Concerning the transport part... The water mass transport and transformation are two related problems. I don't think a simple geostrophy (note a coarse grid in the Irminger Sea and missing profiles in the western Labrador Sea - both are important for budgeting the fluxes) is sufficient for constraining the transports. Frankly, I would not even bring the transport part in the work discussing the contributions of source waters. I think the most important part for now is building a method adequate for the task and thoroughly investigating every aspect of interaction by taking into account a huge baggage of what is known and available to this date and developing something better than a static 2D approach for analysing a strongly time and space variant 3D dynamics and variability – essentially 4D.

*Regarding the velocity field, you are probably right about the Labrador Sea. However, we present in the paper the results of water mass transport across the OVIDE section, which geostrophic velocity field was solved by the box inverse model technique that has been validated by favorable comparisons with independent measurements (Gourcuff et al., 2011; Daniault et al., 2011; 2016; Mercier et al., 2015). The subsampling in the Irminger Basin was taken into consideration by Zunino et al. (2017), who determined that the calculations of the transports through the GEOVIDE radial was robust despite the subsampling of certain regions and concluded that the final errors of the dynamical structures in 2014 are of the same order of magnitude as the errors estimated in previous OVIDE cruises. Regarding the method used to solve the dynamics and variability of the water mass circulation and transformation in the Subpolar North Atlantic, please refer to the answers to your previous comments.*

To conclude my review I share my thinking of this problem as cookbook analogy – all we try to come up here with is a recipe. Think of real ingredients and not those appearing someplace somewhere – if you use the latter, the results are not going to tell much about your true ingredients. On the other hand, by weighting the real properties of the waters with the sought and found fractions, one should come to a section plot similar to that observed.

Considering the amount of data, effort and work needed to address the issues I raised in my review, I recommend rejection. This only reason why it is not revision is that by redoing the paper the authors would come with a totally new method, sets of results and visions. Sorry, but I cannot see it any simpler than that.

*Considering all the arguments compiled in the answers to your comments, we demonstrate that the OMP analysis is a suitable tool to study water mass distributions. We also prove that our choice of SWTs is appropriate to describe all the cruise samples, as evidenced by the low residuals and high correlation coefficients ($R^2$, Table 1) between the measured values (water samples) and those resulting from the mixture of the SWTs. The water mass distribution resulting from our OMP set up is in agreement with the accepted knowledge of the Subpolar North*

*Atlantic circulation. Therefore, we are confident that the submitted work is suitable for its purpose, to provide a framework for interpreting the observed distributions of the trace elements and their isotopes along the GEOVIDE cruise.*

*References*

*Álvarez, M., Pérez, F. F., Bryden, H., and Ríos, A. F.: Physical and biogeochemical transports structure in the North Atlantic subpolar gyre, Journal of Geophysical Research, 109, C03027, doi:10.1029/2003JC002015, 2004.*

*Carracedo, L. I., Pardo, P. C., Flecha, S., and Pérez, F. F.: On the Mediterranean Water Composition, J. Phys. Oceanogr., 46, 1339–1358, doi:10.1175/JPO-D-15-0095.1, 2016.*

*Daniault, N., Lherminier, P., and Mercier, H.: Circulation and transport at the southeast tip of Greenland, Journal of Physical Oceanography, 41 (3), 437–457, doi:10.1175/2010JPO4428.1, 2011.*

*Daniault, N., Mercier, H., Lherminier, P., Sarafanov, A., Falina, A., Zunino, P., Pérez, F. F., Ríos, A. F., Ferron, B., Huck, T., Thierry, V., and Gladyshev, S.: The northern North Atlantic Ocean mean circulation in the early 21st century, Progress in Oceanography, 146, 142–158, doi:10.1016/j.pocean.2016.06.007, 2016.*

*de Jong, M. F., and de Steur,L.: Strong winter cooling over the Irminger Sea in winter 2014–2015, exceptional deep convection, and the emergence of anomalously low SST, Geophysical Research Letters, 43, 7106–7113, doi:10.1002/2016GL069596, 2016.*

*de la Paz, M., García-Ibáñez, M. I., Steinfeldt, R., Ríos, A. F., and Pérez, F. F.: Ventilation versus biology: What is the controlling mechanism of nitrous oxide distribution in the North Atlantic?, Global Biogeochem. Cycles, 31, 745–760, doi:10.1002/2016GB005507, 2017.*

*Dickson, B., Yashayaev, I., Meincke, J., Turrell, B., Dye, S., and Holfort, J.: Rapid freshening of the deep North Atlantic Ocean over the past four decades, Nature, 416 (6883), 832–837, doi:10.1038/416832a, 2002.*

*Fogelqvist, E., Blindheim, J., Tanhua, T., Østerhus, S., Buch, E., and Rey, F.: Greenland-Scotland overflow studied by hydro-chemical multivariate analysis, Deep-Sea Res. Pt. I, 50, 73–102, doi:10.1016/S0967-0637(02)00131-0, 2003.*

*Gebbie, G., and Huybers, P.: Total matrix intercomparison: a method for determining the geometry of water-mass pathways, Journal of Physical Oceanography, 40(8), 1710–1728, doi:10.1175/2010JPO4272.1, 2010.*

*Gourcuff, C., Lherminier, P., Mercier, H., and Le Traon, P. Y.: Altimetry combined with hydrography for ocean transport estimation, Journal of Atmospheric and Oceanic Technology, 28 (10), 1324–1337, doi:10.1175/2011JTECHO818.1, 2011.*

*Henry-Edwards, A., and Tomczak, M.: Remote detection of water property changes from a time series of oceanographic data, Ocean Sci., 2, 11–18, doi:10.5194/os-2-11-2006, 2006.*

*Jeansson, E., Jutterström, S., Rudels, B., Anderson, L. G., Olsson, K. A., Jones, E. P., Smethie, W. M., and Swift, J. H.: Sources to the East Greenland Current and its contribution to the Denmark Strait Overflow, Progress in Oceanography, 78 (1), 12–28, doi:10.1016/j.pocean.2007.08.031, 2008.*

*Jenkins, W. J., Smethie Jr., W. M., Boyle, E .A., and Cutter, G. A.: Water mass analysis for the U.S. GEOTRACES (GA03) North Atlantic sections, Deep Sea Research Part II: Topical Studies in Oceanography, 116, 6–20, doi:10.1016/j.dsr2.2014.11.018, 2015.*

*Lauderdale, J. M., Bacon, S., Naveira Garabato, A. C., and Holliday, N. P.: Intensified turbulent mixing in the boundary current system of southern Greenland, Geophys. Res. Lett., 35, L04611, doi:10.1029/2007GL032785, 2008.*

*Llanillo, P. J., Pelegrí, J. L., Stramma, L.,and Karstensen, J.: ENSO and long-term AAIW shoaling influence on the ventilation of the tropical eastern South Pacific oxygen minimum zone, EUR-OCEANS Conference - A changing ocean, doi:10.13140/2.1.3127.2002, 2013.*

*Mercier, H., Lherminier, P., Sarafanov, A., Gaillard, F., Daniault, N., Desbruyères, D., Falina, A., Ferron, B., Gourcuff, C., and Huck, T.: Variability of the meridional overturning circulation at the Greenland-Portugal OVIDE section from 1993 to 2010, Progress in Oceanography, 132, 250–261, doi:10.1016/j.pocean.2013.11.001, 2015.*

*Pardo, P. C., Pérez, F. F., Velo, A., and Gilcoto, M.: Water masses distribution in the Southern Ocean: Improvement of an extended OMP (eOMP) analysis, Progress in Oceanography, 103, 92–105, doi:10.1016/j.pocean.2012.06.002, 2012.*

*Piron, A., Thierry, V., Mercier, H., and Caniaux, G.: Gyre scale deep convection in the subpolar North-Atlantic Ocean during winter 2014-2015, Geophysical Research Letters, 44(3), 1439–1447, doi:10.1002/2016GL071895, 2017.*

*Rudels, B., Fahrbach, E., Meincke, J., Budéus, G., and Eriksson, P.: The East Greenland Current and its contribution to the Denmark Strait overflow, ICES Journal of Marine Science: Journal du Conseil, 59 (6), 1133–1154, doi:10.1006/jmsc.2002.1284, 2002.*

*Schmitz Jr., W.J., and McCartney, M.S.: On the North Atlantic Circulation, Rev. Geophys., 31(1), 29–49, doi:10.1029/92RG02583, 1993.*

*Sy, A., Rhein, M., Lazier, J. R. N., Koltermann, K. P., Meincke, J., Putzka, A., and Bersch, M.: Surprisingly rapid spreading of newly formed intermediate waters across the North Atlantic Ocean, Nature, 386, 675–679, doi:10.1038/386675a0, 1997.*

*Tanhua, T., Olsson, K. A., and Jeansson, E.: Formation of Denmark Strait overflow water and its hydro-chemical composition, Journal of Marine Systems, 57, 264–288, doi:10.1016/j.jmarsys.2005.05.003, 2005.*

*Tanhua, T., Olsson, K. A., and Jeansson, E.: Tracer evidence of the origin and variability of Denmark Strait Overflow Water. In: Dickson, R.R., Jens, M., Rhines, P. (Eds.), Arctic-Subarctic Ocean Fluxes: Defining the Role of the Northern Seas in Climate. Springer, Science + Business Media B.V., P.O. Box 17, AA Dordrecht, The Netherlands, 475–503, 2008.*

*Xu, X., Schmitz Jr., W. J., Hurlburt, H. E., Hogan, P. J., and Chassignet, E. P.: Transport of Nordic Seas overflow water into and within the Irminger Sea: An eddy-resolving simulation and observations, J. Geophys. Res., 115, C12048, doi:10.1029/2010JC006351, 2010.*

*Yashayaev, I., van Aken, H.M., Holliday, N.P., and Bersch, M.: Transformation of the Labrador Sea Water in the subpolar North Atlantic, Geophys. Res. Lett., 34, L22605, doi:10.1029/2007GL031812, 2007.*

*Yashayaev, I., and Dickson, R. R.: Transformation and fate of overflows in the Northern North Atlantic. In: Dickson, R.R., Jens, M., Rhines, P. (Eds.), Arctic-Subarctic Ocean Fluxes: Defining the Role of the Northern Seas in Climate. Springer, Science + Business Media B.V., P.O. Box 17, AA Dordrecht, The Netherlands, pp. 505–526, 2008.*

*Yashayaev, I., and Loder J. W.: Recurrent replenishment of Labrador Sea Water and associated decadal-scale variability, J. Geophys. Res. Oceans, 121, 8095–8114, doi:10.1002/2016JC012046, 2016.*

*Zunino, P., Lherminier, P., Mercier, H., Daniault, N., García-Ibáñez, M. I., and Pérez, F. F.: The GEOVIDE cruise in May–June 2014 reveals an intense Meridional Overturning Circulation over a cold and fresh subpolar North Atlantic, Biogeosciences, 14, 5323–5342, doi:10.5194/bg-14-5323-2017, 2017.*